# Interaction-Force Transport Gradient Flows

**Egor Gladin**
Humboldt University of Berlin
Berlin, Germany
& HSE University
egorgladin@yandex.ru

**Pavel Dvurechensky**
Weierstrass Institute for
Applied Analysis and Stochastics
Berlin, Germany
pavel.dvurechensky@wias-berlin.de

**Alexander Mielke**
Humboldt University of Berlin
& WIAS
Berlin, Germany
alexander.mielke@wias-berlin.de

**Jia-Jie Zhu**[*]
Weierstrass Institute for
Applied Analysis and Stochastics
Berlin, Germany
jia-jie.zhu@wias-berlin.de

## Abstract

This paper presents a new gradient flow dissipation geometry over non-negative and probability measures. This is motivated by a principled construction that combines the unbalanced optimal transport and interaction forces modeled by reproducing kernels. Using a precise connection between the Hellinger geometry and the maximum mean discrepancy (MMD), we propose the interaction-force transport (IFT) gradient flows and its spherical variant via an infimal convolution of the Wasserstein and spherical MMD tensors. We then develop a particle-based optimization algorithm based on the JKO-splitting scheme of the mass-preserving spherical IFT gradient flows. Finally, we provide both theoretical global exponential convergence guarantees and improved empirical simulation results for applying the IFT gradient flows to the sampling task of MMD-minimization. Furthermore, we prove that the spherical IFT gradient flow enjoys the best of both worlds by providing the global exponential convergence guarantee for both the MMD and KL energy.

## 1 Introduction

Optimal transport (OT) distances between probability measures, including the earth mover's distance [Werman et al., 1985, Rubner et al., 2000] and Monge-Kantorovich or Wasserstein distance [Villani, 2008], are one of the cornerstones of modern machine learning as they allow performing a variety of machine learning tasks, e.g., unsupervised learning [Arjovsky et al., 2017, Bigot et al., 2017], semi-supervised learning [Solomon et al., 2014], clustering [Ho et al., 2017], text classification [Kusner et al., 2015], image retrieval, clustering and classification [Rubner et al., 2000, Cuturi, 2013, Sandler and Lindenbaum, 2011], and distributionally robust optimization [Sinha et al., 2020, Mohajerin Esfahani and Kuhn, 2018]. Many recent works in machine learning adopted the techniques from PDE gradient flows over optimal transport geometries and interacting particle systems for inference and sampling tasks. Those tools not only add new interpretations to the existing algorithms, but also provide a new perspective on designing new algorithms.

---

[*]Corresponding author: Jia-Jie Zhu

38th Conference on Neural Information Processing Systems (NeurIPS 2024).

For example, the classical Bayesian inference framework minimizes the Kulback-Leibler divergence towards a target distribution $\pi$. From the optimization perspective, this can be viewed as solving

$$\min_{\mu \in A \subset \mathcal{P}} \left\{ F(\mu) := \mathrm{D}_{\mathrm{KL}}(\mu|\pi) \right\}, \tag{1}$$

where $A$ is a subset of the space of probability measures $\mathcal{P}$, e.g., the Gaussian family. The Wasserstein gradient flow of the KL gives the Fokker-Planck equation, which can be simulated using the Langevin SDE for MCMC. Beyond the KL, many researchers following Arbel et al. [2019] advocated using the squared MMD instead as the driving energy for the Wasserstein gradient flows for sampling. However, in contrast to the KL setting, there is little sound convergence analysis for the MMD-minimization scheme like the celebrated Bakry-Émery Theorem. Furthermore, it was shown, e.g., in [Korba et al., 2021], that Arbel et al. [2019]'s algorithm suffers a few practical drawbacks. For example, their particles tend to collapse around the mode or get stuck at local minima, and the algorithm requires a heuristic noise injection strategy that is tuned over the iterations; see Figure 1 and §4 for illustrations. Subsequently, many such as Carrillo et al. [2019], Chewi et al. [2020], Korba et al. [2021], Glaser et al. [2021], Craig et al. [2023], Hertrich et al. [2023], Neumayer et al. [2024] proposed modified energies to be used in the Wasserstein gradient flows. In contrast, this paper does not propose

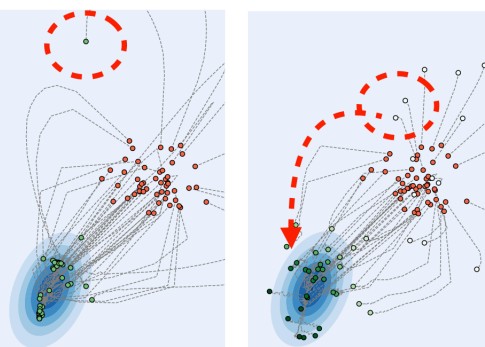

Figure 1: (Left) Wasserstein flow of the MMD energy [Arbel et al., 2019]. Some particles get stuck at points away from the target. (Right) IFT gradient flow (this paper) of the MMD energy. Particle mass is teleported to close to the target, avoiding local minima. Hollow circles indicate particles with zero mass. The red dots are the initial particles, and the green dots are the target distribution. See §4 for more details.

new energy objectives. Instead, we propose a new gradient flow geometry – the IFT gradient flows. To summarize, our main contributions are:

1. We propose the interaction-force transport (IFT) gradient flow geometry over non-negative measures and spherical IFT over probability measures, constructed from the first principles of the reaction-diffusion type equations, previously studied in the context of the Hellinger-Kantorovich (Wasserstein-Fisher-Rao) distance and gradient flows. It was first studied by three groups including Chizat et al. [2018, 2019], Liero et al. [2018], Kondratyev et al. [2016], Gallouët and Monsaingeon [2017]. Our IFT gradient flow is based on the inf-convolution of the Wasserstein and the newly constructed spherical MMD Riemannian metric tensors. This new unbalanced gradient flow geometry allows teleporting particle mass in addition to transportation, which avoids the flow getting stuck at local minima; see Figure 1 for an illustration.

2. We provide theoretical analysis such as the *global exponential decay* of energy functionals via the Polyak-Łojasiewicz type functional inequalities. As an application, we provide the first global exponential convergence analysis of IFT for both the MMD and KL energy functionals. That is, the IFT gradient flow enjoys the best of both worlds.

3. We provide a new algorithm for the implementation of the IFT gradient flow. We then empirically demonstrate the use of the IFT gradient flow for the MMD inference task. Compared to the original MMD-energy-flow algorithm of Arbel et al. [2019], IFT flow does not suffer issues such as the collapsing-to-mode issue. Leveraging the first-principled spherical IFT gradient flow, our method does not require a heuristic noise injection that is commonly tuned over the iterations in practice; see [Korba et al., 2021] for a discussion. Our method can also be viewed as addressing a long-standing issue of the kernel-mean embedding methods [Smola et al., 2007, Muandet et al., 2017, Lacoste-Julien et al., 2015] for optimizing the support of distributions.

**Notation** We use the notation $\mathcal{P}(\boldsymbol{X}), \mathcal{M}^+(\boldsymbol{X})$ to denote the space of probability and non-negative measures on the closed, bounded, (convex) set $\boldsymbol{X} \subset \mathbb{R}^d$. The base space symbol $\boldsymbol{X}$ is often dropped if there is no ambiguity in the context. We note also that many of our results hold for $\boldsymbol{X} = \mathbb{R}^d$. In this paper, the first variation of a functional $F$ at $\mu \in \mathcal{M}^+$ is defined as a function $\frac{\delta F}{\delta \mu}[\mu]$ such that $\frac{\mathrm{d}}{\mathrm{d}\epsilon} F(\mu + \epsilon \cdot v)|_{\epsilon=0} = \int \frac{\delta F}{\delta \mu}[\mu](x) \, \mathrm{d}v(x)$ for any valid perturbation in measure $v$ such that $\mu + \epsilon \cdot v \in \mathcal{M}^+$ when working with gradient flows over $\mathcal{M}^+$ and $\mu + \epsilon \cdot v \in \mathcal{P}$ over $\mathcal{P}$. We often

omit the time index $t$ to lessen the notational burden, e.g., the measure at time $t$, $\mu(t, \cdot)$, is written as $\mu$. The infimal convolution (inf-convolution) of two functions $f, g$ on Banach spaces is defined as $(f \square g)(x) = \inf_y \{f(y) + g(x - y)\}$. In formal calculation, we often use measures and their density interchangeably, i.e., $\int f \cdot \mu$ means the integral w.r.t. the measure $\mu$. For a rigorous generalization of flows over continuous measures to discrete measures, see [Ambrosio et al., 2005]. $\nabla_2 k(\cdot, \cdot)$ denotes the gradient w.r.t. the second argument of the kernel.

## 2 Background

### 2.1 Gradient flows of probability measures for learning and inference

Gradient flows are powerful tools originated from the field of PDE. The intuition can be easily seen from the perspective of optimization as solving the variational problem

$$\min_{\mu \in A \subset \mathcal{M}^+(\boldsymbol{X})} F(\mu)$$

using a "continuous-time version" of gradient descent, over a suitable metric space and, in particular, Riemannian manifold. Since the seminal works by Otto [1996] and colleagues, one can view many PDEs as gradient flows over the aforementioned Wasserstein metric space, canonically denoted as $(\mathcal{P}_2(\boldsymbol{X}), W_2)$; see [Villani, 2008, Santambrogio, 2015] for a comprehensive introduction.

Different from a standard OT problem, a gradient flow solution traverses along the path of the fastest dissipation of the energy $F$ allowed by the corresponding geometry. In this paper, we are only concerned with the geometries with a (pseudo-)Riemannian structure, such as the Wasserstein, (spherical) Hellinger or Fisher-Rao geometries. In such cases, a formal Otto calculus can be developed to greatly simplify the calculations. For example, the Wasserstein Onsager operator (which is the inverse of the Riemannian metric tensor) $\mathbb{K}_W(\rho) : T_\rho^* \mathcal{M}^+ \to T_\rho \mathcal{M}^+, \xi \mapsto -\operatorname{div}(\rho \nabla \xi)$, where $T_\rho \mathcal{M}^+$ is the tangent plane of $\mathcal{M}^+$ at $\rho$ and $T_\rho^* \mathcal{M}^+$ the cotangent plane. Using this notation, a Wasserstein gradient flow equation of some energy $F$ can be written as

$$\dot{\mu} = -\mathbb{K}_W(\mu) \frac{\delta F}{\delta \mu} = \operatorname{div}(\mu \nabla \frac{\delta F}{\delta \mu}). \tag{2}$$

In essence, many machine learning applications are about making different choices of the energy $F$ in (2), e.g., the KL, $\chi^2$-divergence, or MMD. However, Wasserstein and its flow equation (2) are by no means the only meaningful geometry for gradient flows. One major development in the field is the Hellinger-Kantorovich a.k.a. the Wasserstein-Fisher-Rao (WFR) gradient flow. The WFR gradient flow equation is given by the reaction-diffusion equation, for some scaling coefficients $\alpha, \beta > 0$,

$$\dot{u} = \alpha \cdot \operatorname{div}(u \nabla \frac{\delta F}{\delta u}) - \beta u \cdot \frac{\delta F}{\delta u}. \tag{3}$$

A few recent works have applied WFR to sampling and inference [Yan et al., 2024, Lu et al., 2019] by choosing the energy functional to be the KL divergence.

### 2.2 Reproducing kernel Hilbert space and MMD

In this paper, we refer to a bi-variate function $k : \boldsymbol{X} \times \boldsymbol{X} \to \mathbb{R}$ as a symmetric positive definite kernel if $k$ is symmetric and, for all $n \in \mathbb{N}, \alpha_1, \ldots, \alpha_n \in \mathbb{R}$ and all $x_1, \ldots, x_n \in \boldsymbol{X}$, we have $\sum_{i=1}^n \sum_{j=1}^n \alpha_i \alpha_j k(x_j, x_i) \geq 0$. $k$ is a reproducing kernel if it satisfies the reproducing property, i.e., for all $x \in \boldsymbol{X}$ and all functions in a Hilbert space $f \in \mathcal{H}$, we have $f(x) = \langle f, k(\cdot, x) \rangle_{\mathcal{H}}$. Furthermore, the space $\mathcal{H}$ is an RKHS if the Dirac functional $\delta_x : \mathcal{H} \mapsto \mathbb{R}, \delta_x(f) := f(x)$ is continuous. It can be shown that there is a one-to-one correspondence between the RKHS $\mathcal{H}$ and the reproducing kernel $k$. Suppose the kernel is square-integrable $\|k\|_{L_\rho^2}^2 := \int k(x, x) d\rho(x) < \infty$ w.r.t. $\rho \in \mathcal{P}$. The integral operator $\mathcal{T}_{k,\rho} : L_\rho^2 \to \mathcal{H}$ is defined by $\mathcal{T}_{k,\rho} g(x) := \int k(x, x') g(x') d\rho(x')$ for $g \in L_\rho^2$. With an abuse of terminology, we refer to the following composition also as the integral operator

$$\mathcal{K}_\rho := \operatorname{Id} \circ \mathcal{T}_{k,\rho}, \ L^2(\rho) \to L^2(\rho).$$

$\mathcal{K}_\rho$ is compact, positive, self-adjoint, and nuclear; cf. [Steinwart and Christmann, 2008]. To simplify the notation, we simply write $\mathcal{K}$ when $\rho$ is the Lebesgue measure.

The kernel maximum mean discrepancy (MMD) [Gretton et al., 2012] emerged as an easy-to-compute alternative to optimal transport for computing the distance between probability measures, i.e., $\mathrm{MMD}^2(\mu, \nu) := \|\mathcal{K}(\mu - \nu)\|_{\mathcal{H}}^2 = \int \int k(x, x')\, \mathrm{d}(\mu - \nu)(x)\, \mathrm{d}(\mu - \nu)(x')$, where $\mathcal{H}$ is the RKHS associated with the (positive-definite) kernel $k$. While the MMD enjoys many favorable properties, such as a closed-form estimator and favorable statistical properties [Tolstikhin et al., 2017, 2016], its mathematical theory is less developed compared to the Wasserstein space especially in the geodesic structure and gradient flow geometries. It has been shown by Zhu and Mielke [2024] that MMD is a (de-)kernelized Hellinger or Fisher-Rao distance by using a dynamic formulation

$$\mathrm{MMD}^2(\mu, \nu) = \min \left\{ \int_0^1 \|\xi_t\|_{\mathcal{H}}^2\, \mathrm{d}t \,\middle|\, \dot{u} = -\mathcal{K}^{-1}\xi_t, u(0) = \mu, u(1) = \nu,\ \xi_t \in \mathcal{H} \right\}. \quad (4)$$

Mathematically, we can obtain the MMD geodesic structure if we kernelize the Hellinger (Fisher-Rao) Riemannian metric tensor,

$$\mathbb{G}_{\mathrm{MMD}} = \mathcal{K}_\mu \circ \mathbb{G}_{\mathsf{He}}(\mu), \quad \mathbb{K}_{\mathrm{MMD}} = \mathbb{K}_{\mathsf{He}}(\mu) \circ \mathcal{K}_\mu^{-1}, \quad (5)$$

noting that the Onsager operator $\mathbb{K}$ is the inverse of the Riemannian metric tensor $\mathbb{K} = \mathbb{G}^{-1}$. The MMD suffers from some shortcomings in practice, such as the vanishing gradients and kernel choices that require careful tuning; see e.g., [Feydy et al., 2019]. Furthermore, a theoretical downside of the MMD as a tool for optimizing distributions, and kernel-mean embedding [Smola et al., 2007, Muandet et al., 2017] in general, is that they do not allow *transport* dynamics. This limitation is manifested in practice, e.g., it is intractable to optimize the location of particle distributions; see e.g. [Lacoste-Julien et al., 2015]. In this paper, we address all those issues.

## 3   IFT **gradient flows over non-negative and probability measures**

In this section, we propose the IFT gradient flows over non-negative and probability measures. Note that our methodology is fundamentally different from a few related works in kernel methods and gradient flows such as [Arbel et al., 2019, Korba et al., 2021, Hertrich et al., 2023, Glaser et al., 2021, Neumayer et al., 2024] in that we are not concerned with the Wasserstein flows of a different energy, but a new gradient flow dissipation geometry.

### 3.1   (Spherical) IFT **gradient flow equations over non-negative and probability measures**

The construction of the Wasserstein-Fisher-Rao gradient flows crucially relies on the inf-convolution from convex analysis [Liero et al., 2018, Chizat, 2022]. There, the WFR metric tensor is defined using an inf-convolution of the Wasserstein tensor

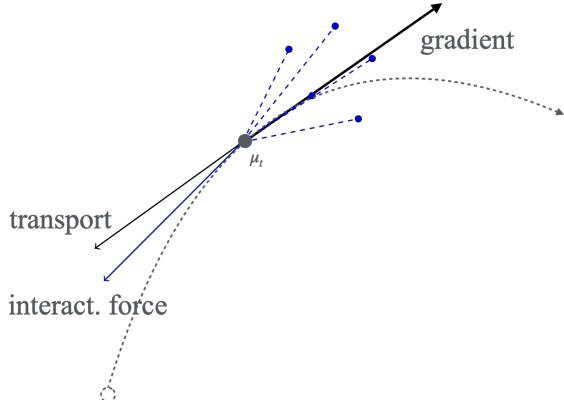

Figure 2: Illustration of the IFT gradient flow. Atoms are subject to both the transport (Kantorovich) potential and the interaction (repulsive) force from other atoms.

and the Hellinger (Fisher-Rao) tensor $\mathbb{G}_{\mathsf{WFR}}(\mu) = \mathbb{G}_W(\mu) \Box \mathbb{G}_{\mathsf{He}}(\mu)$. By Legendre transform, its inverse, the Onsager operator, is given by the sum $\mathbb{K}_{\mathsf{WFR}}(\mu) = \mathbb{K}_W(\mu) + \mathbb{K}_{\mathsf{He}}(\mu)$. Therefore, we construct the IFT gradient flow by replacing the Hellinger (Fisher-Rao) tensor with the MMD tensor, as in (5).

$$\mathbb{G}_{\mathrm{IFT}}(\mu) = \mathbb{G}_W(\mu) \Box \mathbb{G}_{\mathrm{MMD}}(\mu), \quad \mathbb{K}_{\mathrm{IFT}}(\mu) = \mathbb{K}_W(\mu) + \mathbb{K}_{\mathrm{MMD}}(\mu). \quad (6)$$

The MMD gradient flow equation is derived by Zhu and Mielke [2024] using the Onsager operator (5),

$$\dot{\mu} = -\mathbb{K}_{\mathrm{MMD}}(\mu) \frac{\delta F}{\delta \mu}[\mu] = -\mathcal{K}^{-1} \frac{\delta F}{\delta \mu}[\mu]. \quad (7)$$

Hence, we obtained the desired IFT gradient flow equation using (6).

$$\dot{\mu} = -\alpha \mathbb{K}_W(\mu) \frac{\delta F}{\delta \mu}[\mu] - \beta \mathbb{K}_{\mathrm{MMD}}(\mu) \frac{\delta F}{\delta \mu}[\mu] = \alpha \cdot \mathrm{div}(\mu \nabla \frac{\delta F}{\delta \mu}[\mu]) - \beta \cdot \mathcal{K}^{-1} \frac{\delta F}{\delta \mu}[\mu]. \quad (8)$$

Formally, the IFT gradient flow equation can also be viewed as a kernel-approximation to the Wasserstein-Fisher-Rao gradient flow equation, i.e., the reaction-diffusion equation (3).

**Corollary 3.1.** *Suppose $\int k_\sigma(x, \cdot) \, \mathrm{d}\mu = 1$ and the kernel-weighted-measure converges to the Dirac measure $k_\sigma(x, \cdot) \, \mathrm{d}\mu \to \mathrm{d}\delta_x$ as the bandwidth $\sigma \to 0$. Then, the* IFT *gradient flow equation* (8) *tends towards the WFR gradient flow equation as $\sigma \to 0$, i.e., the reaction-diffusion equation* (3).

Like the WFR gradient flow over non-negative measures, the gradient flow equation (8) and (7) are not guaranteed to stay within the probability measure space, i.e., total mass 1. This is useful in many applications such as chemical reaction systems. However, probability measures are often required for machine learning applications. We now provide a mass-preserving gradient flow equation that we term the *spherical* IFT *gradient flow*. The term spherical is used to emphasize that the flow stays within the probability measure, as in the spherical Hellinger distance [Laschos and Mielke, 2019].

To this end, we must first study *spherical MMD* flows over probability measures. Recall that (7) is a Hilbert space gradient flow (see [Ambrosio et al., 2005]) and does not stay within the probability space. Closely related, many works using kernel-mean embedding [Smola et al., 2007, Muandet et al., 2017] also suffer from this issue of not respecting the probability space. To produce a restricted (or projected) flow in $\mathcal{P}$, our starting point is the *MMD minimizing-movement scheme* restricted to the probability space

$$\mu^{k+1} \leftarrow \underset{\mu \in \mathcal{P}}{\mathrm{argmin}} \, F(\mu) + \frac{1}{2\eta} \mathrm{MMD}^2(\mu, \mu^k). \quad (9)$$

We now derive the following mass-preserving spherical gradient flows for the MMD and IFT.

**Proposition 3.2** (Spherical MMD and spherical IFT gradient flow equations)**.** *The spherical MMD gradient flow equation is given by (where $1$ denotes the constant scalar)*

$$\dot{\mu} = -\mathcal{K}^{-1} \left( \frac{\delta F}{\delta \mu}[\mu] - \frac{\int \mathcal{K}^{-1} \frac{\delta F}{\delta \mu}[\mu]}{\int \mathcal{K}^{-1} 1} \right). \quad (10)$$

*Consequently, the spherical* IFT *gradient flow equation is given by*

$$\dot{\mu} = \alpha \cdot \mathrm{div}(\mu \nabla \frac{\delta F}{\delta \mu}[\mu]) - \beta \cdot \mathcal{K}^{-1} \left( \frac{\delta F}{\delta \mu}[\mu] - \frac{\int \mathcal{K}^{-1} \frac{\delta F}{\delta \mu}[\mu]}{\int \mathcal{K}^{-1} 1} \right). \quad (11)$$

*Furthermore, those equations are mass-preserving, i.e., $\int \dot{\mu} = 0$.*

So far, we have identified the gradient flow equations of interest. Now, we are ready to present our main theoretical results on the convergence of the IFT gradient flow via functional inequalities. For example, the logarithmic Sobolev inequality (LSI)

$$\left\| \nabla \log \frac{\mathrm{d}\mu}{\mathrm{d}\pi} \right\|_{L^2(\mu)}^2 \geq c_{\mathrm{LSI}} \cdot \mathrm{D}_{\mathrm{KL}}(\mu | \pi) \text{ for some } c_{\mathrm{LSI}} > 0 \quad (\mathrm{LSI})$$

is sufficient to guarantee the convergence of the pure Wasserstein gradient flow of the KL divergence energy, which governs the same dynamics as the Langevin equation. The celebrated Bakry-Émery Theorem [Bakry and Émery, 1985], is a cornerstone of convergence analysis for dynamical systems as it provides an explicit sufficient condition: suppose the target measure $\pi$ is $\lambda$-log concave for some $\lambda > 0$, then the global convergence is guaranteed, i.e.,

$$\pi = e^{-V} \, \mathrm{d}x \text{ and } \nabla^2 V \geq \lambda \cdot \mathrm{Id} \implies (\mathrm{LSI}) \text{ with } c_{\mathrm{LSI}} = 2\lambda \implies \text{glob. exp. convergence.}$$

The question we answer below is whether the IFT gradient flow enjoys such favorable properties. Our starting point is the (Polyak-)Łojasiewicz type functional inequality.

**Theorem 3.3.** *Suppose the following Łojasiewicz type inequality holds for some $c > 0$,*

$$\alpha \cdot \left\| \nabla \frac{\delta F}{\delta \mu} [\mu] \right\|_{L_\mu^2}^2 + \beta \cdot \left\| \frac{\delta F}{\delta \mu} [\mu] \right\|_{\mathcal{H}}^2 \geq c \cdot \left( F(\mu(t)) - \inf_\mu F(\mu) \right) \tag{IFT-Łoj}$$

*for the* IFT *gradient flow, or*

$$\alpha \cdot \left\| \nabla \frac{\delta F}{\delta \mu} [\mu] \right\|_{L_\mu^2}^2 + \beta \cdot \left\| \frac{\delta F}{\delta \mu} [\mu] - \frac{\int \mathcal{K}^{-1} \frac{\delta F}{\delta \mu} [\mu]}{\int \mathcal{K}^{-1} 1} \right\|_{\mathcal{H}}^2 \geq c \cdot \left( F(\mu(t)) - \inf_\mu F(\mu) \right) \tag{SIFT-Łoj}$$

*for the spherical* IFT *gradient flow. Then, the energy $F$ decays exponentially along the corresponding gradient flow, i.e., $F(\mu(t)) - \inf_\mu F(\mu) \leq e^{-ct} \cdot (F(\mu(0)) - \inf_\mu F(\mu))$.*

To understand a specific gradient flow, one must delve into the detailed analysis of the conditions under which the functional inequalities hold instead of assuming them to hold by default. We provide such analysis for the IFT gradient flows next.

### 3.2  Global exponential convergence analysis

**MMD energy functional**  As discussed in the introduction, the MMD energy has been proposed as an alternative to the KL divergence energy for sampling by Arbel et al. [2019][2], where they assume the access to samples from the target measure $y_i \sim \pi$. However, the theoretical convergence guarantees under the MMD energy is a less-exploited topic. Those authors characterized a local decay behavior under the assumption that the $\mu_t$ must already be close to the target measure $\pi$ for all $t > 0$. The assumptions they made are not only restrictive, but also difficult to check. There has also been no global convergence analysis. For example, Arbel et al. [2019]'s Proposition 2 states that the MMD is non-increasing, which is not equivalent to convergence and is easily satisfied by other flows. The mathematical limitation is that the MMD is in general not guaranteed to be convex along the Wasserstein geodesics. In addition, our analysis also does not require the heuristic noise injection step as was required in their implementation. Mroueh and Rigotti [2020] also used the MMD energy but with a different gradient flow, which has been shown by Zhu and Mielke [2024] to be a kernel-regularized inf-convolution of the Allen-Cahn and Cahn-Hilliard type of dissipation. However, Mroueh and Rigotti [2020]'s convergence analysis is not sufficient for establishing (global) exponential convergence as no functional inequality has been established there. In contrast, we now provide full global exponential convergence guarantees.

We first provide an interesting property that will become useful for our analysis.

**Theorem 3.4.** *Suppose the driving energy is the squared MMD, $F(\mu) = \frac{1}{2} \mathrm{MMD}^2(\mu, \pi)$ and initial datum $\mu_0 \in \mathcal{P}$ is a probability measure. Then, the spherical MMD gradient flow equation* (10) *coincides with the MMD gradient flow equation*

$$\dot{\mu} = -(\mu - \pi), \tag{MMD-MMD-GF}$$

*whose solution is a linear interpolation between the initial measure $\mu_0$ and the target measure $\pi$, i.e.,*

$$\mu_t = e^{-t} \mu_0 + (1 - e^{-t})\pi.$$

*Furthermore, same coincidence holds for the spherical* IFT *and* IFT *gradient flow equation*

$$\dot{\mu} = \alpha \cdot \mathrm{div} \left( \mu \int \nabla_2 k(x, \cdot) \, d(\mu - \pi)(x) \right) - \beta(\mu - \pi). \tag{MMD-IFT-GF}$$

The explicit solution to the ODE (MMD-MMD-GF) shows an exponential convergence to the target measure $\pi$ along the (spherical) MMD gradient flow. The (spherical) IFT gradient flow equation (MMD-IFT-GF) differs from the Wasserstein gradient flow equation of [Arbel et al., 2019] by a linear term. This explains the intuition of why we can expect good convergence properties for the IFT gradient flow of the squared MMD energy. We exploit this feature of the IFT gradient flow to show global convergence guarantees for inference with the MMD energy. This has not been possible previously when confined to the pure Wasserstein gradient flow.

---

[2]We do not use the terminology "MMD gradient flow" from [Arbel et al., 2019] since it is inconsistent with the naming convention of "Wasserstein gradient flow" as Wasserstein refers to the geometry, not the functional.

**Theorem 3.5** (Global exponential convergence of the IFT flow of the MMD energy). *Suppose the energy $F$ is the squared MMD energy $F(\mu) = \frac{1}{2} \mathrm{MMD}^2(\mu, \nu)$. Then, the* (IFT-Łoj) *holds globally with a constant $c \geq 2\beta > 0$.*

*Consequently, for any initialization within the non-negative measure cone $\mu_0 \in \mathcal{M}^+$, the squared MMD energy decays exponentially along the IFT gradient flow of non-negative measures, i.e.,*

$$\frac{1}{2} \mathrm{MMD}^2(\mu_t, \nu) \leq e^{-2\beta t} \cdot \frac{1}{2} \mathrm{MMD}^2(\mu_0, \nu). \tag{12}$$

*Furthermore, if the initial datum $\mu_0$ and the target measure $\pi$ are probability measures $\mu_0, \pi \in \mathcal{P}$, then the squared MMD energy decays exponentially globally along the spherical IFT gradient flow, i.e., the decay estimate* (12) *holds along the spherical IFT gradient flow of probability measures.*

We emphasize that no Bakry-Émery type or kernel conditions are required – the Łojasiewicz inequality holds globally when using the IFT flow. In contrast, the Wasserstein-Fisher-Rao gradient flow

$$\dot{\mu} = \alpha \cdot \mathrm{div} \left( \mu \int \nabla_2 k(x, \cdot) \, \mathrm{d}\,(\mu - \pi)\,(x) \right) - \beta \cdot \int k(x, \cdot) \, \mathrm{d}\,(\mu - \pi)\,(x) \quad \text{(MMD-WFR-GF)}$$

does not enjoy such global convergence guarantees.

**Global exponential convergence under the KL divergence energy**  For variational inference and MCMC, a common choice of the energy is the KL divergence energy, i.e., $F(\mu) = \mathrm{D}_{\mathrm{KL}}(\mu|\pi)$. This has already been studied by a large body of literature, including the case of Wasserstein-Fisher-Rao [Liero et al., 2023, Lu et al., 2019]. Not surprisingly, (LSI) is still sufficient for the exponential convergence of the WFR type of gradient flows since the dissipation of the Wasserstein part alone is sufficient for driving the system to equilibrium. For the IFT gradient flows under the KL divergence energy functional, the convergence can still be established. This showcases the strength of the IFT geometry – it enjoys the best of both worlds. The IFT gradient flow equation of the KL divergence energy reads $\dot{\mu} = \alpha \cdot \mathrm{div}(\mu \nabla \log \frac{\mathrm{d}\mu}{\mathrm{d}\pi}) - \beta \cdot \mathcal{K}^{-1} \log \frac{\mathrm{d}\mu}{\mathrm{d}\pi}$. Unlike the MMD-energy flow case, the spherical IFT gradient flow of the KL over probability measures $\mathcal{P}$ no longer coincides with that of the (non-spherical) IFT and is given by

$$\dot{\mu} = \alpha \cdot \mathrm{div}(\mu \nabla \log \frac{\mathrm{d}\mu}{\mathrm{d}\pi}) - \beta \cdot \mathcal{K}^{-1} \left( \log \frac{\mathrm{d}\mu}{\mathrm{d}\pi} - \frac{\int \mathcal{K}^{-1} \log \frac{\mathrm{d}\mu}{\mathrm{d}\pi}}{\int \mathcal{K}^{-1} 1} \right). \tag{13}$$

**Proposition 3.6** (Exponential convergence of the SIFT gradient flow of the KL divergence energy). *Suppose the* (LSI) *holds with $c_{LSI} = 2\lambda$ or the target measure $\pi$ is $\lambda$-log concave for some $\lambda > 0$. Then, the KL divergence energy decays exponentially globally along the spherical IFT gradient flow* (13) *, i.e., $\mathrm{D}_{\mathrm{KL}}(\mu_t|\pi) \leq e^{-2\alpha\lambda t} \mathrm{D}_{\mathrm{KL}}(\mu_0|\pi)$.*

The intuition behind the above result is that the SIFT gradient flow converges whenever the pure Wasserstein gradient flow, i.e., its convergence is at least as fast as the Wasserstein gradient flow. However, we emphasize that the decay estimate of the KL divergence energy only holds along the spherical IFT flow over probability measures $\mathcal{P}$, but not the full IFT flow over non-negative measures $\mathcal{M}^+$.

### 3.3 Minimizing movement, JKO-splitting, and a practical particle-based algorithm

In applications to machine learning and computation, continuous-time flow can be discretized via the JKO scheme [Jordan et al., 1998], which is based on the minimizing movement scheme (MMS) [De Giorgi, 1993]. For the reaction-diffusion type gradient flow equation in the Wasserstein-Fisher-Rao setting, the *JKO-splitting* a.k.a. *time-splitting* scheme has been studied by Gallouët and Monsaingeon [2017], Mielke et al. [2023]. This amounts to splitting the diffusion (Wasserstein) and reaction (MMD) step in (8), i.e., at time step $\ell \geq 1$

$$\mu^{\ell + \frac{1}{2}} \leftarrow \underset{\mu \in \mathcal{P}}{\mathrm{argmin}} \, F(\mu) + \frac{1}{2\tau} W_2^2(\mu, \mu^\ell), \qquad \text{(Wasserstein step)}$$

$$\mu^{\ell+1} \leftarrow \underset{\mu \in \mathcal{P}}{\mathrm{argmin}} \, F(\mu) + \frac{1}{2\eta} \mathrm{MMD}^2(\mu, \mu^{\ell + \frac{1}{2}}). \quad \text{(MMD step)}$$

$$\tag{14}$$

A similar JKO-splitting scheme can also be constructed via the WFR gradient flow, which amounts to replacing the MMD step in (14) with a proximal step in the KL (as an approximation to the Hellinger), i.e., $\mu^{\ell+1} \leftarrow \operatorname{argmin}_{\mu \in \mathcal{P}} F(\mu) + \frac{1}{\eta} D_{KL}(\mu|\mu^{\ell+\frac{1}{2}})$, which is well-studied in the optimization literature as the entropic mirror descent [Nemirovskij and Yudin, 1983]. Our MMD step can also be viewed as a mirror descent step with the mirror map $\frac{1}{2}\|\mathcal{K} \cdot \|_{\mathcal{H}}^2$. However, for the task of MMD inference of [Arbel et al., 2019], WFR flow does not possess convergence guarantees such as our Theorem 3.4. The MMD step can also be easily implemented as in our simulation.

We summarize the resulting overall IFT particle gradient descent from the JKO splitting scheme in Algorithm 1 in the appendix. We now look at those two steps respectively. For concreteness, we consider a flexible particle approximation to the probability measures, with possibly non-uniform weights allocated to the particles, i.e., $\mu = \sum_{i=1}^n \alpha_i \delta_{x_i}, \alpha \in \Delta^n, x_i \in \boldsymbol{X}$.

**Wasserstein step: particle position update.** (14) is a standard JKO scheme; see, e.g., [Santambrogio, 2015]. The optimality condition of the Wasserstein proximal step can be implemented using a particle gradient descent algorithm

$$x_i^{\ell+1} = x_i^\ell - \tau \cdot \nabla \frac{\delta F}{\delta \mu}[\mu^\ell](x_i^\ell), \ i = 1, ..., n, \tag{15}$$

which is essentially the algorithm proposed by Arbel et al. [2019] when $F(\mu) = \frac{1}{2}\operatorname{MMD}^2(\mu, \pi)$.

**MMD step: particle weight update.** The MMD step in (14) is a discretization step of the spherical MMD gradient flow, as shown in (9) and Proposition 3.2. We propose to use the updated particle location $x_i^{\ell+1}$ from the Wasserstein step (15) and update the weights $\beta_i$ by solving

$$\inf_{\beta \in \Delta^n} F(\sum_{i=1}^n \beta_i \delta_{x_i^{\ell+1}}) + \frac{1}{2\eta}\operatorname{MMD}^2(\sum_{i=1}^n \beta_i \delta_{x_i^{\ell+1}}, \sum_{i=1}^n \alpha_i^\ell \delta_{x_i^{\ell+1}}),$$

i.e., the MMD step only updates the weights. Alternatively, as in the classical mirror descent optimization, one can use a linear approximation $F(\mu) \approx F(\mu^\ell) + \langle \frac{\delta F}{\delta \mu}[\mu^\ell], \mu - \mu^\ell \rangle_{L^2}$. We also provide a specialized discussion on the MMD-energy minimization task of [Arbel et al., 2019]. Let the energy objective be the squared MMD $F(\mu) := \frac{1}{2}\operatorname{MMD}(\mu, \pi)^2$. In this setting, we are given the particles sampled from the target measure $y^i \sim \pi$. For the MMD step in (14), the computation is drastically simplified to an MMD barycenter problem, which was also studied in [Cohen et al., 2021]. This amounts to solving a convex quadratic program with a simplex constraint; see the appendix for the detailed expression.

## 4 Numerical Example

The overall goal of the numerical experiments is to approximate the target measure $\pi$ by minimizing the squared MMD energy, i.e., $\min_{\mu \in A \subset \mathcal{P}} \operatorname{MMD}^2(\mu, \pi)$. In all the experiments, we have access to the target measure $\pi$ in the form of samples $y_i \sim \pi$. This setting was studied in [Arbel et al., 2019] as well as in many deep generative model applications. In the following experiments, we compare the performance of our proposed algorithm of IFT gradient flow, which implements the JKO-splitting scheme in (14) and is detailed in Algorithm 1, to that of *(1)* Arbel et al. [2019]'s the "MMD flow" (see our discussion in [2]), we used their algorithm both with and without a heuristic noise injection suggested by those authors; *(2)* the Wasserstein-Fisher-Rao flow of the MMD (MMD-WFR-GF). The WFR flow was also used by Yan et al. [2024], Lu et al. [2023] but for minimizing the KL divergence function. As discussed in §3.2, the MMD flow of [Arbel et al., 2019] does not possess global convergence guarantees while IFT does. Furthermore, in the Gaussian mixture target experiment, the target measure $\pi$ is not log-concave. We emphasize that our convergence guarantee still holds for the IFT flow while there is no decay guarantee for the WFR flow. We provide the code for the implementation at `https://github.com/egorgladin/ift_flow`.

**Gaussian target in 2D experiment** Figures 3(a) and 4 showcase the performance of the algorithms in a setting where $\mu^0$ and $\pi$ are both Gaussians in 2D. Specifically, $\mu^0 \sim \mathcal{N}(5 \cdot \boldsymbol{1}, I)$ and $\pi \sim \mathcal{N}\left(\boldsymbol{0}, \left(\begin{smallmatrix} 1 & 1/2 \\ 1/2 & 2 \end{smallmatrix}\right)\right)$. The number of samples drawn from $\mu^0$ and $\pi$ was set to $n = 100$. A Gaussian

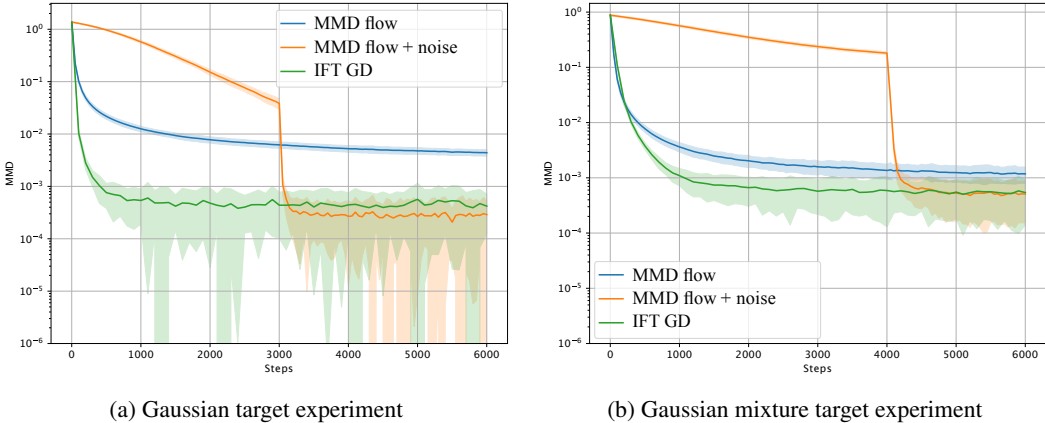

(a) Gaussian target experiment

(b) Gaussian mixture target experiment

Figure 3: Mean loss and standard deviation computed over 50 runs

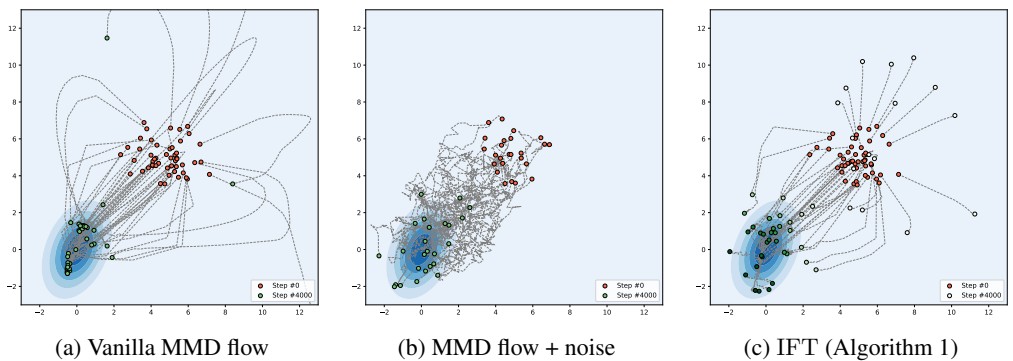

(a) Vanilla MMD flow

(b) MMD flow + noise

(c) IFT (Algorithm 1)

Figure 4: Trajectory of a randomly selected subsample produced by different algorithms in the Gaussian target experiment. Color intensity indicates points' weights. The hollow dots indicate the particles that have already vanished.

kernel with bandwidth $\sigma = 10$ was used. For all three algorithms, we chose the largest stepsize that didn't cause unstable behavior, $\tau = 50$. The parameter $\eta$ in (23) was set to 0.1. As can be observed from the trajectories produced by MMD flow (Figure 4(a)), most points collapse into small clusters near the target mode. Some points drift far away from the target distribution and get stuck; the resulting samples represent the target distribution poorly, which is a sign of suboptimal solution. MMD flow with the heuristic noise injection produces much better results. We suspect the noise

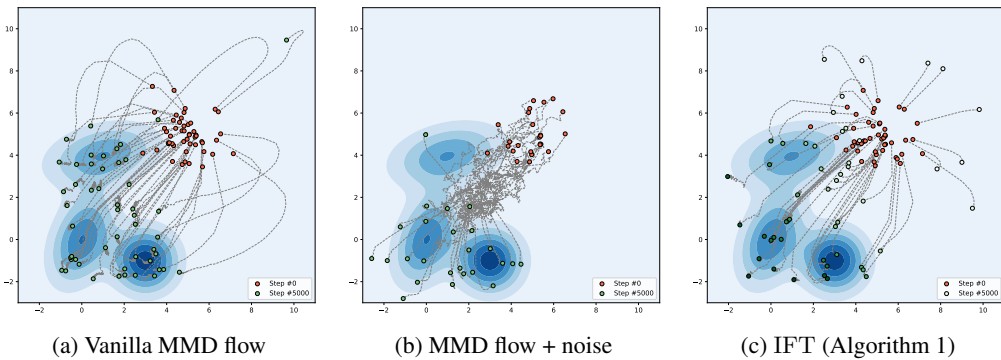

(a) Vanilla MMD flow

(b) MMD flow + noise

(c) IFT (Algorithm 1)

Figure 5: Trajectory of a randomly selected subsample produced by different algorithms in the Gaussian mixture experiment. Color intensity indicates points' weights. The hollow dots indicate the particles that have already vanished.

helps to escape local minima; however, the injection needs to be heuristically tuned. However, it takes a large number iterations for points to get close to locations with high density of the target distribution. Similarly to the previous research on noisy MMD flow, we use a relatively large noise level (10) in the beginning and "turn off" the noise after a sufficient number of iterations (3000 in our case). A drawback of this approach is that the right time for noise deactivation depends on the particular problem instance, which makes the algorithm behavior less predictable.

Algorithm 1 achieves a similar accuracy to that of the noise-injected MMD flow, but much faster – already after 1000 steps – without any heuristic noise injection. For the few particles that did not make it close to the target Gaussian's mean, their mass is teleported to those particles that are close to the target. Hence, the resulting performance of the IFT algorithm does not deteriorate. The hollow dots in the trajectory plot indicate the particles whose mass has been teleported and hence their weights are zero. This is a major advantage of unbalanced transport for dealing with local minima. For a faster implementation, in the implementation of the MMD step in (14), we only perform a single step of projected gradient descent instead of computing a solution to the auxiliary optimization problem (23). To

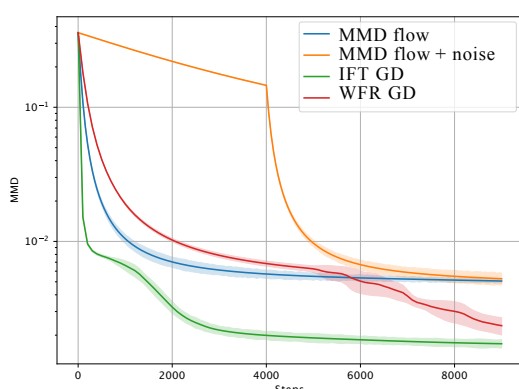

Figure 6: Comparison with the WFR flow of the MMD in 100 dimensions

be fair in comparison, we count each iteration as two steps. Thus, 6000 steps of the algorithm in Figure 3(a) correspond to only 3000 iterations, i.e., the results for IFT algorithm have already been handicapped by a factor of 2. We would also like to note that Algorithm 1 was executed with constant hyperparameters without further tuning over the iterations, in contrast to the noisy MMD flow. In practical implementations, it is possible to further improve the performance by sampling new locations (particle rejuvenation) in the MMD step (14) as done similarly in [Dai et al., 2016]. Since this paper is a theoretical one and not about competitive benchmarking, we leave this for future work.

**Gaussian mixture in 2D experiment** The second experiment has a similar setup. However, this time the target is a mixture of equally weighted Gaussian distributions,

$$\mathcal{N}\left(\mathbf{0}, \begin{pmatrix} 1 & 1/2 \\ 1/2 & 2 \end{pmatrix}\right), \quad \mathcal{N}\left(\begin{pmatrix} 3 \\ -1 \end{pmatrix}, I\right), \quad \mathcal{N}\left(\begin{pmatrix} 1 \\ 4 \end{pmatrix}, \begin{pmatrix} 3 & 1/2 \\ 1/2 & 1 \end{pmatrix}\right).$$

Figures 3(b) and 5 showcase loss curves and trajectories produced by the considered algorithms.

**WFR flow for Gaussian mixture target in 100D** We conducted an experiment in dimension $d = 100$, comparing the IFT flow with the WFR flow of the MMD energy. The initial distribution is $\mathcal{N}(0, I)$, and the target $\pi$ is a mixture of 3 distributions $\mathcal{N}(m_i, \Sigma_i)$, $i = 1, 2, 3$, where $m_i$ and $\Sigma_i$ are randomly generated such that $\|m_i\|_2 = 20$ and the smallest eigenvalue of $\Sigma_i$ is greater than $0.5$. For fairness, each iteration of IFT particle GD (as well as its version with KL step) counts as two steps, i.e., these methods only performed 4500 iterations. In the noisy MMD flow, the noise is disabled after 4000 steps. All methods are used with equal stepsize.

## 5   Discussion

In summary, the (spherical) IFT gradient flows are a suitable choice for energy minimization of both the MMD and KL divergence energies with sound global exponential convergence guarantees. There is also an orthogonal line of works studying the Stein gradient flow and descent [Liu and Wang, 2019, Duncan et al., 2019], which also has the mechanics interpretation of repulsive forces. It can be related to our work in that the IFT gradient flow has a (de-)kernelized reaction part, while the Stein flow has a kernelized diffusion part. Furthermore, there is a work [Manupriya et al., 2024] that proposes a static MMD-regularized Wasserstein distance, which should not be confused with our IFT gradient flow geometry. Another future direction is sampling and inference when we do not have access to the samples from the target distribution $\pi$, but can only evaluate its score function $\nabla \log \pi$.

## Acknowledgments and Disclosure of Funding

We thank Gabriel Peyré for the helpful comments regarding the practical algorithms for the JKO-splitting scheme. This project has received funding from the Deutsche Forschungsgemeinschaft (DFG, German Research Foundation) under Germany's Excellence Strategy – The Berlin Mathematics Research Center MATH+ (EXC-2046/1, project ID: 390685689) and from the priority programme "Theoretical Foundations of Deep Learning" (SPP 2298, project number: 543963649). During part of the project, the research of E. Gladin was prepared within the framework of the HSE University Basic Research Program.

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

# A  Appendix: proofs and additional technical details

*Proof of Proposition 3.2.* We first consider the MMS step (9). The Lagrangian of the MMS step is, for $\lambda \in \mathbb{R}$,

$$\mathcal{L}(\mu, \lambda) = F(\mu) + \frac{1}{2\eta}\text{MMD}^2(\mu, \mu^\ell) + \lambda\left(\int \mu - 1\right).$$

The Euler-Lagrange equation gives

$$\frac{\delta F}{\delta \mu}[\mu^\ell] + \frac{1}{\eta}\mathcal{K}(\mu - \mu^\ell) + \lambda = 0, \tag{16}$$

$$\int \mu = 1. \tag{17}$$

Rewriting the first equation,

$$\mu = \mu^\ell - \eta\mathcal{K}^{-1}\frac{\delta F}{\delta \mu}[\mu^\ell] - \eta\lambda\mathcal{K}^{-1}.$$

Integrating both sides, we obtain

$$1 = 1 - \eta\int \mathcal{K}^{-1}\frac{\delta F}{\delta \mu}[\mu^\ell] - \eta\lambda\int \mathcal{K}^{-1}1 \implies \lambda = -\frac{\int \mathcal{K}^{-1}\frac{\delta F}{\delta \mu}[\mu^\ell]}{\int \mathcal{K}^{-1}1}.$$

Let the time step $\eta \to 0$ in the first equation in the Euler-Lagrange equation (17), we obtain

$$\dot{\mu} = -\mathcal{K}^{-1}\left(\frac{\delta F}{\delta \mu}[\mu] + \lambda\right) = -\mathcal{K}^{-1}\frac{\delta F}{\delta \mu}[\mu] + \frac{\int \mathcal{K}^{-1}\frac{\delta F}{\delta \mu}[\mu]}{\int \mathcal{K}^{-1}1} \cdot \mathcal{K}^{-1}1, \tag{18}$$

which is the desired spherical MMD gradient flow equation.

Spherical IFT gradient flow equation is obtained by an inf-convolution [Gallouët and Monsaingeon, 2017, Liero et al., 2018, Chizat et al., 2019] of the above spherical MMD and Wasserstein part. The verification of the mass-preserving property is by a straightforward integration of (18)

$$0 = \int \dot{\mu} = -\int \mathcal{K}^{-1}\frac{\delta F}{\delta \mu}[\mu] + \frac{\int \mathcal{K}^{-1}\frac{\delta F}{\delta \mu}[\mu]}{\int \mathcal{K}^{-1}1} \cdot \int \mathcal{K}^{-1}1 = 0.$$

Hence, the theorem is proved. $\qquad\square$

*Proof of Theorem 3.3.* The proof amounts to identifying the correct left-hand side of the Łojasiewicz type inequality. We take the time derivative of the energy

$$\frac{d}{dt}F(\mu) = \langle\frac{\delta F}{\delta \mu}[\mu], \alpha \cdot \text{div}(\mu\nabla\frac{\delta F}{\delta \mu}[\mu]) - \beta \cdot \mathcal{K}^{-1}\frac{\delta F}{\delta \mu}[\mu]\rangle_{L^2}$$

$$= -\alpha \cdot \|\nabla\frac{\delta F}{\delta \mu}[\mu]\|_{L^2_\mu}^2 - \beta \cdot \|\frac{\delta F}{\delta \mu}[\mu]\|_{\mathcal{H}}^2,$$

which is the desired left-hand side of the Łojasiewicz type inequality.

For the spherical IFT gradient flow, we have

$$\frac{d}{dt}F(\mu) = \langle\frac{\delta F}{\delta \mu}[\mu], \alpha \cdot \text{div}(\mu\nabla\frac{\delta F}{\delta \mu}[\mu]) - \beta \cdot \mathcal{K}^{-1}\left(\frac{\delta F}{\delta \mu}[\mu] - \frac{\int \mathcal{K}^{-1}\frac{\delta F}{\delta \mu}[\mu]}{\int \mathcal{K}^{-1}1}\right)\rangle_{L^2}$$

$$= \langle\frac{\delta F}{\delta \mu}[\mu], \alpha\cdot\text{div}(\mu\nabla\frac{\delta F}{\delta \mu}[\mu])\rangle_{L^2} + \langle\frac{\delta F}{\delta \mu}[\mu] - \frac{\int \mathcal{K}^{-1}\frac{\delta F}{\delta \mu}[\mu]}{\int \mathcal{K}^{-1}1}, -\beta\cdot\mathcal{K}^{-1}\left(\frac{\delta F}{\delta \mu}[\mu] - \frac{\int \mathcal{K}^{-1}\frac{\delta F}{\delta \mu}[\mu]}{\int \mathcal{K}^{-1}1}\right)\rangle_{L^2}$$

$$= -\alpha \cdot \left\|\nabla\frac{\delta F}{\delta \mu}[\mu]\right\|_{L^2_\mu}^2 - \beta \cdot \left\|\frac{\delta F}{\delta \mu}[\mu] - \frac{\int \mathcal{K}^{-1}\frac{\delta F}{\delta \mu}[\mu]}{\int \mathcal{K}^{-1}1}\right\|_{\mathcal{H}}^2,$$

where the second equality follows from the fact that the spherical IFT gradient flow is mass-preserving. Hence, the left-hand side of the Łojasiewicz type inequality is obtained.

$\qquad\square$

*Proof of Corollary 3.1.* The formal proof is by using well-known results for kernel smoothing in non-parametric statistics [Tsybakov, 2009]. Note that the gradient flow equation can be rewritten as

$$\dot{u} = -\alpha \cdot \text{div}(\mu \nabla \frac{\delta F}{\delta \mu}[\mu]) + \beta \cdot \mathcal{K}^{-1} \frac{\delta F}{\delta \mu}[\mu] = -\alpha \cdot \text{div}(\mu \nabla \frac{\delta F}{\delta \mu}[\mu]) + \beta \cdot \mu \mathcal{K}_\mu^{-1} \frac{\delta F}{\delta \mu}[\mu].$$

Recall the definition of the integral operator

$$\mathcal{K}f = \int k_\sigma(\cdot, y) f(y) \, \mathrm{d}y, \quad \mathcal{K}_\mu f = \int k_\sigma(\cdot, y) f(y) \, \mathrm{d}\mu(y). \tag{19}$$

Formally, as $k_\sigma(x, \cdot) \, \mathrm{d}\mu \to \mathrm{d}\delta_x$, we have $\mathcal{K}_\mu \xi \to \xi$ for any $\xi \in L^2(\mu)$. Then, the IFT gradient flow equation tends towards the PDE

$$\dot{\mu} = -\alpha \cdot \text{div}(\mu \nabla \frac{\delta F}{\delta \mu}[\mu]) + \beta \mu \cdot \frac{\delta F}{\delta \mu}[\mu]. \tag{20}$$

Furthermore, we also have $\|\frac{\delta F}{\delta \mu}[\mu]\|_\mathcal{H}^2 \to \|\frac{\delta F}{\delta \mu}[\mu]\|_{L^2(\mu)}^2$. Hence, the conclusion follows. $\square$

*Proof of Theorem 3.4.* We first recall that the first variation of the squared MMD energy is given by

$$\frac{\delta}{\delta \mu}\left(\frac{1}{2}\text{MMD}^2(\mu, \nu)\right)[\mu] = \int k(x, \cdot)(\mu - \nu)(\, \mathrm{d}x). \tag{21}$$

Plugging the first variation of the squared MMD energy into the gradient flow equation (7), (10), (8), and (11), we obtain the desired flow equations in the theorem. The ODE solution is obtained by an elementary argument. $\square$

*Proof of Theorem 3.5.* We take the time derivative of the energy and apply the chain rule formally and noting the gradient flow equation (8),

$$\frac{\mathrm{d}}{\mathrm{d}t} F(\mu) = \langle \int \frac{\delta F}{\delta \mu}[\mu], \alpha \cdot \text{div}(\mu \nabla \frac{\delta F}{\delta \mu}[\mu]) - \beta \cdot \mathcal{K}^{-1} \frac{\delta F}{\delta \mu}[\mu] \rangle_{L^2}$$

$$= -\alpha \cdot \|\nabla \frac{\delta F}{\delta \mu}[\mu]\|_{L_\mu^2}^2 - \beta \cdot \|\frac{\delta F}{\delta \mu}[\mu]\|_\mathcal{H}^2 \leq -\beta \cdot \|\frac{\delta F}{\delta \mu}[\mu]\|_\mathcal{H}^2.$$

Plugging in $F(\mu) = \frac{1}{2}\text{MMD}^2(\mu, \nu)$, an elementary calculation shows that

$$\frac{\mathrm{d}}{\mathrm{d}t}\left(\frac{1}{2}\text{MMD}^2(\mu, \nu)\right) \leq -\beta \cdot \|\int k(x, \cdot)(\mu - \nu)(\, \mathrm{d}x)\|_\mathcal{H}^2 = -2\beta \cdot \frac{1}{2}\text{MMD}^2(\mu, \nu), \tag{22}$$

which establishes the desired Łojasiewicz inequality specialized to the squared MMD energy, which reads

$$\alpha \cdot \left\|\nabla \int k(x, \cdot)(\mu - \nu)(\, \mathrm{d}x)\right\|_{L_\mu^2}^2 + \beta \cdot \left\|\int k(x, \cdot)(\mu - \nu)(\, \mathrm{d}x)\right\|_\mathcal{H}^2 \geq c \cdot \frac{1}{2}\text{MMD}^2(\mu, \nu). \tag{Łoj}$$

By Grönwall's lemma, exponential decay is established.

Furthermore, plugging the first variation of the squared MMD energy into the gradient flow equation (10), the extra term in the spherical flow equation becomes

$$\frac{\int \mathcal{K}^{-1} \frac{\delta F}{\delta \mu}[\mu]}{\int \mathcal{K}^{-1} 1} = \frac{\int \mathcal{K}^{-1} \mathcal{K}(\mu - \pi)}{\int \mathcal{K}^{-1} 1} = 0.$$

Hence, the coincidence is proved. $\square$

*Proof of Proposition 3.6.* By the Bakry-Émery Theorem, we have the LSI (LSI) hold with $c_{\text{LSI}} = 2\lambda$. Taking the time derivative of the KL divergence energy along the SIFT gradient flow, we have

$$\frac{\mathrm{d}}{\mathrm{d}t}D_{\text{KL}}(\mu_t | \pi) = \left\langle \nabla \log \frac{\mathrm{d}\mu_t}{\mathrm{d}\pi}, \dot{\mu}_t \right\rangle_{L^2}$$

$$= -\alpha \cdot \left\|\nabla \log \frac{\mathrm{d}\mu_t}{\mathrm{d}\pi}\right\|_{L_{\mu_t}^2}^2 - \beta \cdot \left\|\log \frac{\mathrm{d}\mu_t}{\mathrm{d}\pi} - \frac{\int \mathcal{K}^{-1} \log \frac{\mathrm{d}\mu_t}{\mathrm{d}\pi}}{\int \mathcal{K}^{-1} 1}\right\|_\mathcal{H}^2$$

$$\overset{\text{(LSI)}}{\leq} -2\alpha\lambda \cdot D_{\text{KL}}(\mu_t | \pi) + 0.$$

By Grönwall's lemma, exponential convergence is established. $\square$

Note that this result does not hold for the full IFT flows over non-negative measures as there exists no LSI globally on $\mathcal{M}^+$.

**Remark A.1** (Regularized inverse of the integral operator). *Strictly speaking, the integral operator $\mathcal{K}$ is compact and hence its inverse is unbounded. Using a viscosity-regularization techniques by Efendiev and Mielke [2006], we can obtain the flow equation where the inverse is always well-defined, i.e., $\dot{\mu} = \alpha \cdot \mathrm{div}(\mu \nabla \frac{\delta F}{\delta \mu}[\mu]) - \beta \cdot (\mathcal{K} + \epsilon \cdot I)^{-1} \frac{\delta F}{\delta \mu}[\mu]$. This corresponds to an additive regularization of the kernel Gram matrix in practical computation.*

**A particle gradient descent algorithm for** IFT **gradient flows**   We use the notation $K_{XX}$ to denote the kernel Gram matrix $K_{XX} = [k(x_i^{\ell+1}, x_j^{\ell+1})]_{i,j=1}^n$, $K_{X\bar{X}}$ for the cross kernel matrix $K_{X\bar{X}} = [k(x_i^{\ell+1}, x_j^\ell)]_{i,j=1}^n$, etc.

---

**Algorithm 1** A JKO-splitting for IFT particle gradient descent

---

**Require:**
1: **for** $\ell = 1$ to $T - 1$ **do**
2:    Compute the first variation of the energy $F$ at $\mu^\ell$: $g^\ell = \frac{\delta F}{\delta \mu}[\mu^\ell]$. Then,

$$x_i^{\ell+1} \leftarrow x_i^\ell - \tau^\ell \cdot \nabla g^\ell(x_i^\ell), \quad i = 1, ..., n \qquad \text{(Wasserstein step)}$$

$$\alpha^{\ell+1} \leftarrow \underset{\alpha \in \Delta^n}{\mathrm{argmin}}\, F(\sum_{i=1}^n \alpha_i \delta_{x_i^{\ell+1}}) + \frac{1}{2\eta^\ell} \begin{bmatrix} \alpha \\ \alpha^\ell \end{bmatrix}^\top \begin{pmatrix} K_{XX} & -K_{X\bar{X}} \\ -K_{X\bar{X}} & K_{\bar{X}\bar{X}} \end{pmatrix} \begin{bmatrix} \alpha \\ \alpha^\ell \end{bmatrix} \quad \text{(MMD step)}$$

3: **end for**
4: Output the particle measure $\widehat{\mu}^T = \sum_{i=1}^n \alpha_i^T \delta_{x_i^T}$.

---

**Implementing the MMD step**   The MMD step in the JKO-splitting scheme is a convex quadratic program with a simplex constraint, which can be formulated as

$$\inf_{\beta \in \Delta^n} \frac{1}{2} \mathrm{MMD}^2(\sum_{i=1}^n \beta_i \delta_{x_i^{\ell+1}}, \pi) + \frac{1}{2\eta} \mathrm{MMD}^2(\sum_{i=1}^n \beta_i \delta_{x_i^{\ell+1}}, \sum_{i=1}^n \alpha_i^\ell \delta_{x_i^{\ell+1}}). \qquad (23)$$

We further expand the optimization objective (multiplied by a factor of $2\tau$ for convenience)

$$\tau \cdot \left\| \sum_{i=1}^n \beta_i \phi(x_i^{k+1}) - \frac{1}{m} \sum_{j=1}^m \phi(y_j) \right\|^2 + \left\| \sum_{i=1}^n \beta_i \phi(x_i^{k+1}) - \sum_{i=1}^n \alpha_i^k \phi(x_i^k) \right\|^2$$

$$= \tau \cdot \left( \beta^\top K_{XX} \beta - \frac{2}{m} \beta^\top K_{XY} \mathbf{1} + \frac{1}{m^2} \mathbf{1}^\top K_{YY} \mathbf{1} \right) + \left( \beta^\top K_{XX} \beta - 2\beta^\top K_{X\bar{X}} \alpha + \alpha^\top K_{\bar{X}\bar{X}} \alpha \right)$$

$$= (1 + \tau) \beta^\top K_{XX} \beta - \frac{2\tau}{m} \beta^\top K_{XY} \mathbf{1} - 2\beta^\top K_{X\bar{X}} \alpha + \frac{\tau}{m^2} \mathbf{1}^\top K_{YY} \mathbf{1} + \alpha^\top K_{\bar{X}\bar{X}} \alpha. \quad (24)$$

Therefore, the MMD step in Algorithm 1 can be implemented as a convex quadratic program with a simplex constraint.

**A particle gradient descent algorithm for the WFR flow of the MMD energy**   We provide the implementation details of the WFR flow of the MMD energy. The goal is to simulate the PDE (MMD-WFR-GF). To the best of our knowledge, there has been no prior implementation of this flow. Nor is there a convergence guarantee. Similar to the JKO-splitting scheme of the IFT flow in (14), (MMD-WFR-GF) can be discretized using the two-step scheme

$$\mu^{\ell+\frac{1}{2}} \leftarrow \arg\min_{\mu \in \mathcal{P}} F(\mu) + \frac{1}{2\tau} W_2^2(\mu, \mu^\ell) \quad \text{(Wasserstein step)}$$

$$\mu^{\ell+1} \leftarrow \underset{\mu \in \mathcal{P}}{\mathrm{argmin}}\, F(\mu) + \frac{1}{\eta} \mathrm{KL}(\mu, \mu^{\ell+\frac{1}{2}}) \quad \text{(KL step)}$$

where the energy function $F$ is the squared MMD energy, $F(\mu) = \frac{1}{2} \mathrm{MMD}^2(\mu, \pi)$. Use the explicit Euler scheme, the KL step amounts to the entropic mirror descent. In the optimization literature, this

step can be implemented as multiplicative update of the weights (or density), i.e., suppose $x_i^{\ell+1}$ is the new particle location after the Wasserstein step, then we update the weights vector $\alpha$ via

$$\alpha_i^{\ell+1} \leftarrow \alpha_i^{\ell} \cdot \exp\left(-\eta \cdot \frac{\delta F}{\delta \mu}[\mu^{\ell}](x_i^{\ell+1})\right).$$

