# OpenReview forum: "Interaction-Force Transport Gradient Flows"
_NeurIPS.cc/2024/Conference — NeurIPS 2024 poster_

### Official Review · Reviewer_qhY4 · 2024-06-28

**Soundness:** 3
**Presentation:** 2
**Contribution:** 2
**Rating:** 4
**Confidence:** 3

**Summary:**

The paper proposes a gradient flow in combined Wasserstein-MMD geometry w.r.t. certain functionals. The authors primarily consider MMD squared functional, but also have some theoretical results regarding KL divergence functional. The work is more about theory: the authors are concerned about some mathematical properties of their proposed flows and convergence analysis, and have only toy 2D illustrative experiments.

**Strengths:**

* Overall, the considered topic is quite interesting. The theory of Gradient Flows in different geometries is an emergent field which is at the intersection of Machine (Deep) learning and mathematics. This theory is full of remarkable, non-trivial, theoretical results. Transmitting all of this mathematical beauty into practical algorithms is praiseworthy.
* The paper has some interesting theoretical results and statements.

**Weaknesses:**

* (A) At first, I found the manuscript to be a bit difficult to read, especially section 2. A lot of specific mathematical terms were used, e.g., “Onsager operator”; tangent/cotangent spaces and metric tensor of (probability) measures space. A lot of relationships between these specific objects were mentioned, e.g., $\mathbb{K} = \mathbb{G}^{-1}$; formulation of gradient flow through the Onsager operator (eq. 2). I think that in order to make the text more accessible for those who are not a specialist in geometry of (probability) measure spaces, it should be either simplified, or all necessary theoretical introductions should be done, e.g., in the appendix.
* (B) I am not fully satisfied with the structure of the text. In particular:
    * Why Remark 3.4 and Corollary 3.5. (some properties of pure IFT gradflow which was introduced much earlier) are located right after technical Theorem 3.3 (Lojasiewicz inequalities)? I think it is better to place these statements right after Remark 3.1.
    * For me, it is a bit strange that the paper develops theory of spherical IFT gradflows (Section 3.1.), while the only practically considered case (where the driving functional for the gradflow is MMD) does not require this theory (Theorem 3.6) because spherical MMD (IFT) coincides with conventional MMD (IFT) flow. May be more emphasize (inculuding practical evaluations) should be put on KL-driving gradflows, where the sphericity matters.
* (C) (lines 98-99). Machine learning applications of Wasserstein gradient flows: some missed links: [1-6]
* (D) To be honest, I am a bit skeptical about the pure MMD gradient flow (in the MMD geometry) - which is denoted as (MMD-MMD-GF) in Theorem 3.6, and, correspondingly, my skepticism extends to MMD gradflow in the joined Wasserstein-MMD geometry - (MMD-IFT-GF) in Theorem 3.6. At first, (MMD-MMD-GF) was considered in literature, e.g., [7] (not cited!) - see Section 3.1, Case 1 of their paper. And it was noted that such pure MMD-MMD flow is undesirable in practice, exactly because it “teleports” mass between initial and target distribution (note that the solution to MMD-MMD is just interpolation between distributions, as noted by Theorem 3.6). As I understand, the idea of the paper under consideration is that by considering joined Wasserstein-MMD geometry (MMD-IFT flow) one can leverage this problem. However, in the paper, I didn’t find sufficient evidence that it is the case. In particular, the proposed practical optimization procedure includes solving proximal MMD minimizing-movement step, eq. 16. For $F = \text{MMD}$ it boils down to eq. (18), which is MMD barycenter problem. It is known that MMD barycenter problem has solution, see [8, proposition 2], - it is just a mixture of input distributions. Therefore, if we fairly solve MMD minimizing movement step, the resulting $\mu^{\ell + 1}$ will mix $\mu^{\ell + \frac{1}{2}}$ and target $\pi$, i.e., the teleporting of mass will occur.
* (E) The practical validation of the method is rather weak. Only a couple of 2D experiments with Gaussians/Mixture of Gaussians. Moreover, I didn’t find that the proposed method performs better than the alternatives. May be, according to some metrics it is indeed the case, but the visual performance of the method is somewhat disappointing. As I understand from Figures 3, 4 and gifs provided in the supplementary, the method leaves a considerable number of points far from the support of target distribution. The alternatives, even vanilla MMD flow, are better in terms of this characteristic.

[1] Gao et. al., Deep Generative Learning via Variational Gradient Flow, ICML’2019

[2] Gao et. al, Deep Generative Learning via Euler Particle Transport, MSML’2021

[3] Mokrov et. al., Large-Scale Wasserstein Gradient Flows, NeurIPS’2021

[4] Alvarez-Melis et. al., Optimizing Functionals on the Space of Probabilities with Input Convex Neural Networks, TMLR’2022

[5] Bunne et. al., Proximal Optimal Transport Modeling of Population Dynamics, AISTATS’2022

[6] Fan et. al., Variational Wasserstein gradient flow, ICML’2022

[7] Mroueh et. al., Sobolev Descent, AISTATS’2019

[8] Cohen et. al., Estimating Barycenters of Measures in High Dimensions

**Questions:**

* (a) What are the properties of inverse operator $\mathcal{K}^1$. In particular, why is it linear on its arguments?
* (b) The MMD minimizing movement step (eq. 16 and eq. 18) is being solved inexactly in practice, e.g., only weights of particles are optimized, while the exact solution is the mixture of source and target distributions. Moreover, even eq. 18 is substituted with a single step of project GD. What is the reason? How do such approximations affect theoretical and practical properties of the proposed method?
* (c) In the appendix, proofs. Why does the equality hold: $\langle \frac{\delta F}{\delta \mu}[\mu], \mathcal{K}^{-1}\frac{\delta F}{\delta \mu}[\mu] \rangle_{L_\mu^2} = \Vert \frac{\delta F}{\delta \mu}[\mu] \Vert^2_{\mathcal{H}}$?

**Limitations:**

The limitations were addressed correctly

---

> ### Author Rebuttal · Authors · 2024-08-04
>
> We thank the reviewer for writing a long and critical review. We must point out that the review is filled with misunderstandings, which we do our best to clarify below. We would appreciate it if the reviewer could please consider our clarification.
>
> >  This theory is full of remarkable, non-trivial, theoretical results. Transmitting all of this mathematical beauty into practical algorithms is praiseworthy.
>
> Thank you. But such a positive assessment does **not** match your score.
>
> > (E) The practical validation is rather weak... ... visual performance of the method is somewhat disappointing... the method leaves a considerable number of points far from the support of target distribution... vanilla MMD flow, are better
>
> We are deeply confused by this assessment. The only explanation we can think of is that the reviewer completely misunderstood the point of our experiment and intuition of unbalanced transport. We have clearly stated **twice** in the manuscript (Fig.3 & 4 captions),  that **color intensity indicates points’ weights** and mention that **weights vanish**. The "points far from the support" the reviewer was referring to are likely (since we are confused by the review) the points of zero mass, hence the reviewer's comment is likely due to a major misunderstanding. Nonetheless, we will improve the presentation further to avoid misunderstandings.
> We will explicitly say: **The hollow/white circles indicate the particles that have already vanished**.
>
> Please note: as the paper indicates, the performance of the IFT is overwhelmingly good. Hence the review comment does not make sense.
>
> > only a couple of 2D experiments
>
> We have now included results in higher dimensions in the attached PDF.
>
> > strange that the paper develops theory of spherical IFT gradflows.. does not require this theory (Theorem 3.6) because spherical MMD (IFT) coincides with conventional MMD (IFT) flow
>
> The reasoning seems flawed here. Our results such as Thm 3.6, precisely establish this special-case equivalence of SIFT and allow an easy implementation. Without our theory, there is no literature on the PDE of a mass-preserving flow in the MMD/IFT, then no implementation is possible.
>
> The reviewer claimed our theory is "not needed" after she/he read our theory. Without this result, one cannot implement the mass-preserving flow. Furthermore, rigorously speaking, it is also incorrect to say two gradient flows are equivalent. Hence, the logic of the claim that our result is ”not needed” is not sound
>
> > [7] (not cited!)
>
> We have now cited a subsequent/expanded work by the same group [Y. Mroueh and M. Rigotti] in the revised preprint. This paper covers and overlaps with the older paper you mentioned, we believe the coverage is now enough.
>
> Furthermore, the paper you mentioned did not contain the PDE theory and principled gradient structure we uncovered. Hence, although related, it should not be used as an argument to undermine our contribution.
>
> > my skepticism extends to MMD gradflow in the joined Wasserstein-MMD geometry
>
> Unfortunately, we failed to find sound reasoning for this "extension" of skepticism. We never claimed MMD-Mmd-Gf (which is not our focus) is superior. IFT is what we advocate and the Wasserstein contribution made the difference. The review simply cited an old paper related to but didn't theoretically study **MMD-MMD-GF**, let alone IFT, and claimed this "extends" their "skepticism" to IFT. We do not see any sound reasoning here.
>
> > Therefore, if we fairly solve MMD minimizing movement step, the resulting μℓ+1 will mix μℓ+12 and target π, i.e., the teleporting of mass will occur.
>
> Sorry, we did not get what is the question or concern here. The statement is straightforward from our PDE/ODE characterization in Thm 3.6.
>
> > (a) What are the properties of inverse operator 𝐾1.  In particular, why is it linear
>
> Do we understand it correctly, that the reviewer is asking why is the inverse of a linear operator, linear? Please see also the first paragraph in Sec 2.2 for properties of the integral operator and references that can provide more basics of kernel methods and the integral operator. Or we didn't understand your question correctly?
>
>
> > (c) In the appendix, proofs. Why does the equality hold:
>
> First, there is a typo that the inner products should be in unweighted L2 (thank you for helping us fix this), i.e.,
> $$
> \langle
>        \frac{\delta F}{\delta \mu}\left[\mu\right]
>        ,\mathcal K^{-1}
>        \frac{\delta F}{\delta \mu}\left[\mu\right]
>             \rangle_{L^2}
>             \\
>             =
>  \| \frac{\delta F}{\delta \mu}\left[\mu\right]\|^2_{\cal H}
> $$
> This correct relation follows from the textbook definition of the RKHS (and its norm). See standard text such as
> Cucker, F., & Zhou, D. X. (2007). _Learning theory: an approximation theory viewpoint_ (Vol. 24).
>
> > substituted with a single step of project GD. What is the reason? How do such approximations affect theoretical and practical properties of the proposed method?
>
>
> In optimization, it is often desirable and significantly faster to perform inexact iterations to be more computationally efficient. That is the reason of our implementation. As our paper does not focus on the theory of the exact-inexact relation, hence it does not affect the existing analysis of the paper. In practice, we have observed such changes do not affect the performance significantly but improve the speed. Since eq (18) is a convex program, inexact iterations are preferred.
>
>
> > difficult to read...A lot of specific mathematical terms were used
>
> First, we will consider your suggestions.
>
> Our paper is a mathematically rigorous treatment, hence some mathematical terms are necessary. Plus, we have already opted for a minimal set of common terminologies for ML researchers on the topic. It also seems the other reviewers did not experience the same difficulty. Nonetheless, we will try to make the paper more accessible to non-experts (than it already is).

---

> > ### Comment · Reviewer_qhY4 · 2024-08-12
> > **Thanks to the authors**
> >
> > I thank the authors for the answers they provided and appreciate the fairly expressed attitude towards my review. Some comments:
> >
> > 1. Indeed, I missed that your method supports the weights on par with the particles itself, and this is the reason of my wrong evaluation of your 2D Gaussian experiments. My bad, my carelessness. It seems that I was a bit biased due to typical particle flows I know (MMD flow, KSD flow, SVGD flow) which do not introduce this additional complexity with weights.
> >
> > 2. I appreciate the additional 100D Gaussian $\rightarrow$ mixture of 3 Gaussians experiment. It strengthens the work.
> >
> > 3. **The reviewer claimed our theory is "not needed"** - I never said that anywhere in my review. The only my "not needed" was about ethics review. Regarding the theory, I just wanted to encourage the discussion on the practical aspects of the flows different from MMD (IFT).
> >
> > 4. Which particular work do you mean by **[Y. Mroueh and M. Rigotti]**? Also, I am a little confused as to why the authors refuse to cite a related work [7], which I pointed out in my review.
> >
> > 5. For sure, MMD-MMD-GF is not your focus. My point was that MMD-IFT-GF (the only practically evaluated flow you propose) inherits some undesirable properties of the pure MMD-MMD flow. I mentioned the teleporting of mass problem noticed in the old paper I cited. And I just claimed that this teleporting of mass problem also appears in your case (MMD-IFT-GF) - theoretically - when solving (MMD minimising movement step) - eq. 16. This is because the MMD minimizing movement step (as you noticed) boils down to MMD barycenter problem, which has a known solution (see [8, proposition 2]). And this known solution is just a mixture of source and target samples. Maybe this teleporting of mass phenomenon is indeed clear from your PDE/ODE characterization in Thm 3.6, but anyway this phenomenon worth to be explicitly mentioned in the paper
> >
> > 6. **In optimization, it is often desirable and significantly faster to perform inexact iterations to be more computationally efficient.** In general, I agree with this statement. However in your case solving the MMD barycenter problem exactly seems to be faster, because, as [8, proposition 2] notices, the solution is just the mixture of particles.
> >
> > In conclusion, I thank the authors one more time and raise my score.

---

> > > ### Author Response · Authors · 2024-08-12
> > > **Thank you for acknowledging the major misunderstanding. We have now addressed the new points.**
> > >
> > > We thank the reviewer for reading our rebuttal. However, the 6 points raised by the reviewer appear to digress from the main point of the rebuttal and do not justify the rejection assessment. We now point-by-point address the reviewer's comments. Due to the lateness of the comments, we try our best to be thorough.
> > >
> > > ### Points 1 and 2
> > >
> > > Thank you for acknowledging the major misunderstanding and acknowledging our new experiments. We believe those issues are now resolved.
> > >
> > > ### Point 3
> > >
> > > > Under item (B) of "Weaknesses", it was stated that "the only practically considered case (where the driving functional for the gradflow is MMD) **does not require this theory** (Theorem 3.6)".
> > >
> > > First, we apologize for wrongly writing "required" as "needed", though we believe the meaning is the same. This is what our "not needed" comment refers to. Does the reviewer's claim "not require" does not imply "not need", or does the reviewer still stand by this assessment? In any case, we have already addressed this (non)-issue in the rebuttal, and we believe this point is now resolved.
> > >
> > > ### Point 4
> > >
> > > [Y. Mroueh and M. Rigotti]: Unbalanced sobolev descent. Advances in Neural Information Processing Systems. 2020;33:17034-43.
> > >
> > > > I am a little confused as to why the authors refuse to cite a related work [7]
> > >
> > > We have clearly stated in the rebuttal that the newer paper above contains the old framework but additionally newer methodologies (e.g. Kernel-Sobolev-Fisher discrepancy, which is more general) and results. Proper scholarly practice is to avoid block citations of many similar papers, when the relevant line of work has already been covered. We also need more time to look into the content of the 7 papers [1-7] the reviewer suggested we cite. [Y. Mroueh and M. Rigotti] appears to be more recent and comprehensive to the best of our knowledge. We hope our reason is clear.
> > >
> > > In any case, we also did not refuse to cite [7], we simply stated that we have covered kernel Sobolev descent, and it should not be used as an argument to undermine our contribution. Furthermore, none of those papers contain the contributions of our paper, so we again wish to emphasize that comments digress from the main point of the rebuttal and the rejection assessment is not justified here.
> > >
> > > ### Point 5
> > >
> > > > My point was that MMD-IFT-GF ...  inherits some undesirable properties of the pure MMD-MMD flow.
> > >
> > > We still do not see any mathematical justification for this "inheritance". The comments kept mentioning the MMD steps, but the IFT has a Wasserstein step with diffusion. So the comment is not sound. More mathematical analysis and evidence are needed to support such a claim.
> > >
> > > > anyway this phenomenon worth to be explicitly mentioned in the paper
> > >
> > > We agree. We have already done this by giving the precise mathematical formulation of the flow solution: see the second formula in Thm 3.6. This precise statement is already explicit. Furthermore, the solution of (MMD-IFT-GF) has not been studied and is not simply a mixture. We are open to adding more plain English sentences to the presentation if it helps non-experts understand the results better.
> > >
> > > But again, the reviewer digresses and this is a minor presentation (non-)issue.
> > >
> > > > known solution is just a mixture of source and target samples
> > >
> > > Mathematically, this is not rigorous. We do not assume the target distribution to be discrete, the mixture is infinite-dimensional and not simple to implement. The goal of IFT or Arbel et al.'s work is to find a gradient-based algorithm to generate the path $\mu_t$.
> > >
> > > Again, we do not see this to be a mathematical reason to undermine IFT.
> > >
> > > ### Point 6
> > >
> > > We agree that there can be more discussion and future work on how to implement the MMD step. We already discussed with great detail. We will improve the presentation.
> > >
> > > > solving the MMD barycenter problem exactly seems to be faster
> > > We have tried both in practice. What the reviewer described is not the case. "Faster" for what? We will expand on this in the corresponding section.
> > >
> > > > because, as [8, proposition 2] notices, the solution is just
> > >
> > > But our goal is not to solve the MMD Barycenter sub-problem -- it is just a subroutine in the JKO splitting scheme.
> > > The goal is to generate samples to construct the path $\mu_t$. The logic in the reviewer's comment is flawed: why not just take samples from $\pi$ directly? Why do we bother using methods such as that from [Arbel et al.]?
> > >
> > > Furthermore, [8, proposition 2] simply says the solution to the sub-step is a reweighting, that is precisely what we implemented. One must optimize the reweighting coefficients $\beta_p$ in [8]. The reviewer's comments seem to have trivialized the this.
> > >
> > > Again, zooming out to the big picture, this numerical detail of implementation, for which we did not hide anything, does not seem to be a sound case for rejecting our contribution.
> > >
> > >
> > > ### Conclusion
> > > In summary, we appreciate the reviewer's time and effort. However, those comments do not justify the rejection assessment.

---

> ### Author Response · Authors · 2024-08-12
> **End of discussion period approaching**
>
> Dear reviewer,
>
> Thank you for your feedback on our manuscript. We have carefully considered your comments and suggestions and have made the revisions. We have also included new experiments and done our best to answer your questions. The rebuttal took a tremendous amount of effort and we want to make sure it has been read.
>
> As the discussion period will be closed soon, we kindly ask for your feedback on the rebuttal. Have we addressed your concerns? Is there anything else we can improve?
>
> Thank you again for your time and effort.
>
> Authors

---

### Official Review · Reviewer_f71K · 2024-07-11

**Soundness:** 4
**Presentation:** 4
**Contribution:** 4
**Rating:** 8
**Confidence:** 2

**Summary:**

This manuscript proposes a new gradient flow over probability and non-negative measures termed the interaction-force transport (IFT). The flow is based on the inf-convolution of the Wasserstein Riemannian metric tensor and the spherical maximum mean discrepancy (MMD) Riemannian metric tensor. The authors provide a number of theoretical results related to their proposal: global exponential convergence guarantees for the MMD and Kullback-Leibler divergence energies. While convergence results are available for the KL divergence energy, not much is known theoretically in the context of the MMD energy. The authors then introduce a particle gradient descent algorithm for IFT, composed of two steps gradient flows (one to update particle locations via the Wasserstein step and one to update particle weights via the MMD step), and show two proof-of-concept examples that validate the developed theory.

**Strengths:**

1. The paper is well-written and well-organized. Despite heavy notation throughout, the authors do a nice job introducing the notation and staying consistent throughout the manuscript. The literature review on related method is also quite thorough.
2. In my opinion, this is a significant theoretical contribution to the machine/statistical learning literature. The topic is timely and will appeal to a broad NeurIPS audience.
3. While the paper is theoretical in nature, I appreciate that the authors provide an implementation of the proposed gradient flows. The presented algorithm is easy to understand ensuring reproducibility.
4. The contribution of this manuscript is original in many aspects: the new definition of gradient flows, the proofs of global convergence for the MMD and KL energies, and comparison to previously defined methods for the MMD flow that required a heuristic noise injection step.

**Weaknesses:**

I did not identify many weaknesses in this work. While I understand that the contributions are theoretical in nature, I wish the authors would have presented more examples/comparisons to the work of Arbel et al. [2019] in an appendix. The presented examples are sufficient as proof of concept.

**Questions:**

Could the authors comment on the roughness of the standard deviation bands in Figure 2 for the proposed method? Also, while this limitation is already mentioned briefly in the discussion, I feel that a bit more could be said about scalability of the presented algorithm. This does not necessarily have to be included in the main body of the paper, but rather in the appendix where the algorithm is presented.

**Limitations:**

Limitations were adequately addressed.

---

> ### Author Rebuttal · Authors · 2024-08-04
>
> Thank you for taking the time to carefully read and review the paper. We appreciate that you have noticed the many subtle features we put into the manuscript. Thank you for the kind summary in the "Strengths" section. Below, we respond to a few concerns and suggestions.
>
> > the roughness of the standard deviation bands in Figure 2 for the proposed method?
>
> Thank you for the careful read and for noticing the details.
>
> First, we believe the comment refers to the std band in Fig.2, which occurs around and below 10^{-3} magnitude. We suspect that could be due to that we reported in the y-axis in log scale. Then, it is expected the bands get rougher the lower they get. We observe that the same happens to the "MMD flow+noise" when its performance improves eventually (over more iterations); so it does not pertain to IFT.
>
>
> > this limitation is already mentioned briefly in the discussion, I feel that a bit more could be said about scalability of the presented algorithm. This does not necessarily have to be included in the main body of the paper, but rather in the appendix where the algorithm is presented.
>
> Indeed, we agree that scalability is important and deserves a more detailed discussion, at least in the technical details in the appendix.
>
> First, we note that our current implementation only looks at the batch case. Here, IFT performs very well as in, e.g., Fig.2. As we have larger data sets in higher dimensions, the vanilla MMD might be limiting.
>
> Additionally, to speed up the implementation and hence scalability, we have reported that we perform a single step of projected gradient descent instead of computing a full solution to the optimization problem (18). This is effective as (18) is a convex program.
>
> We note that we have not yet explored stochastic algorithms; we have focused on full-batch so far. Hence stochastic algorithms and parallel implementation could be future directions to improve scalability.
>
> Since we are also interested in applications to generative models, there are possibilities such as deep-net features inside the kernels, as used in MMD-GAN works.
>
>
> > more examples/comparisons to the work of Arbel et al. [2019] in an appendix.
>
> Thank you for the suggestion -- we agree. We have now added new experiments, whose results we report in the PDF:
> - experiments in higher dimensions, vs Arbel et al. 2019;
> - new experiments that uses the Wasserstein-Fiher-Rao flow of the MMD energy. To the best of our knowledge, this is the first implementation of the MMD energy in this flow;
> - many miscellaneous improvements in terms of scalability and implementation, as suggested by the review. We will provide the details in the appendix of the revised manuscript as you suggested.

---

> ### Comment · Reviewer_f71K · 2024-08-08
>
> After considering all of the reviews and the authors' rebuttal, I am inclined to keep my rating as is. I appreciate the additional experiments provided in the PDF file as part of the rebuttal.

---

> > ### Author Response · Authors · 2024-08-12
> > **Thank you for reading the rebuttal**
> >
> > Dear reviewer,
> >
> > Thank you for taking the time to read and respond to our rebuttal! Your feedback has helped improve our manuscript.
> >
> > Authors

---

### Official Review · Reviewer_2Ho7 · 2024-07-12

**Soundness:** 2
**Presentation:** 3
**Contribution:** 2
**Rating:** 5
**Confidence:** 2

**Summary:**

This paper proposes a novel gradient flow geometry – interaction-force transport (IFT). It is theoretically shown that IFT gradient flow has global exponential convergence guarantees both for MMD and KL energies. The authors propose an algorithm based on the JKO-splitting scheme and test it on examples with 2D Gaussians and Gaussian mixture.

**Strengths:**

The paper provides the proof of the exponential convergence guarantees for IFT gradient flow with MMD energy.

**Weaknesses:**

I am not convinced that the established theoretical results and provided experimental justifications are sufficiently significant. The main theoretical contribution of the current paper is the proof of exponential convergence guarantees for their IFT gradient flow both with MMD and KL divergence energy. (Actually, the proofs of these results do not seem to be very impressive, e.g., the proof of Proposition 3.8 immediately follows from two well-known facts. Anyway, it is not my major concern.) The authors state that the established convergence guarantees (especially, for the MMD energy) are the main motivation for considering the IFT gradient flow.  However, for the KL case, it is not a surprising property since even ordinary Wasserstein flow with KL divergence energy has the same exponential convergence guarantees. For the MMD case, the results are quite novel, although there exist several other works which prove some convergence properties of flows with MMD energy (Arbel et al., 2019). Thus, I am wondering, are the provided proofs of exponential convergence rates actually important for the practical use of the designed algorithm? The empirical evaluation of the algorithms seems to be very limited. The algorithm is tested only in low-dimensional experiments using 2D Gaussians and Gaussian Mixtures which immediately raises questions regarding the scalability of the approach. Besides, the authors compare their approach only with Wasserstein flows with MMD energy (with or w/o noise injection) (Arbel et al., 2019, Korba et al., 2021). However, it is important to see how the algorithm behaves in comparison to flows with KL divergence energy as well.

Overall, my main concerns are related to the questionable significance of paper results. The proof of exponential convergence rates for their IFT gradient flow solely does not seem to be a significant contribution. Meanwhile, the experimental evaluation needs to be considerably enhanced.

*Minor*:

- line 218: 'expontial' - typo

**Questions:**

Does your approach have some practical use cases?

I suggest the authors to improve the experimental part of their paper by
- including experiments in dimensions larger than $d=2$
- performing comparisons with flows using KL divergence as the energy, e.g., with (Yan et al., 2023, Lu et al., 2019) which are cited in the paper

I am open to adjusting my score if the authors address these suggestions.

**References.**

M. Arbel, A. Korba, A. Salim, and A. Gretton. Maximum Mean Discrepancy Gradient Flow. arXiv:1906.04370, Dec. 2019.

A. Korba, P.-C. Aubin-Frankowski, S. Majewski, and P. Ablin. Kernel Stein Discrepancy Descent. In Proceedings of the 38th International Conference on Machine Learning, pages 5719–5730. PMLR, July 2021.

Y. Yan, K. Wang, and P. Rigollet. Learning Gaussian Mixtures Using the Wasserstein-Fisher-Rao Gradient Flow. arXiv:2301.01766, Jan. 2023.

Y. Lu, J. Lu, and J. Nolen. Accelerating Langevin Sampling with Birth-death. ArXiv, May 2019.

**Limitations:**

The authors have addressed the limitations of their approach in the discussion section.

---

> ### Author Rebuttal · Authors · 2024-08-04
>
> Thank you for the constructive suggestions and the critical review. We have incorporated most of your suggestions, e.g., new experiments. We did find the one comparison you suggested with KL inference difficult, for which we explained the reason.
>
> > including experiments in dimensions larger than 𝑑=2
>
> Done. The results (attached PDF) are consistent with the other experiments.
>
> > performing comparisons...  (Yan et al., 2023, Lu et al., 2019)
>
> We agree that more experiments can strengthen the paper. We have run new experiments and reported them in the attached PDF, such as WFR flows used by Yan et al. and Lu et al.
>
> First, We must note that KL minimization is NOT the topic of this paper: our focus is a type of (unbalanced-transport) gradient flows and optimization with applications to the MMD-minimization tasks. IFT is tailored to address the gap in the literature on "MMD flow". We also do not claim that MMD inference is superior to KL inference, while some other papers might do so. Furthermore, we look forward to applying our theory to generative models using MMD, e.g., [Galashov et al].
>
> Nonetheless, we appreciate the reviewer raising this. We can think of two interpretations of the question:
>
> 1. KL energy with the proposed IFT flow:
> Recall that the MMD step is a discretization of the differential equation
> $$
> \dot \mu =- \beta\cdot \operatorname{\mathcal{K}}^{-1}
>     \left(\log \frac{\mathrm{d}\mu}{\mathrm{d}\pi} - ... \right)
> $$
> One could view the update step as a step in the "kernel-mean-embedding space": let $e(\mu):=\operatorname{\mathcal{K}} \mu$  be the kernel mean embedding
> $$
> e(\mu)^{\ell+1}
> \gets
> e(\mu)^{\ell}  - \eta \beta \cdot \left(\log \frac{\mathrm{d}\mu^{\ell}}{\mathrm{d}\pi} -...\right)
> $$
> However, as there is no guarantee that the velocity $\log \frac{\mathrm{d}\mu^{\ell}}{\mathrm{d}\pi}-...$ to be in the RKHS. Hence, this step is theoretically interesting, but it's unclear how the infinite-dimensional update can be implemented in a principled way.
>
> 2. KL energy with the WFR flow [Yan & Lu et al.]:
> Their methods are adaptations of the original Wasserstein-Fisher-Rao flow. Their cases belong to the "KL-inference" where one only has access to the score (or potentials) of the target e.g. $\nabla \log \pi$, instead of the "MMD-inference" (our paper, and Arbel et al., etc.) where we have access to the target $\pi$ via its samples but not the score $\nabla \log \pi$. Therefore, how to perform a "fair comparison" is unclear to us at the moment (e.g. how many samples from $\pi$ ? Noisy evaluation of the score?).
>
> In summary, we agree that the reviewer's proposal is interesting and can see room for future work. At this moment, we can only leave comparing MMD vs KL as future work as they apply in different tasks and settings (score-based vs sample-based).
>
> We have now implemented the **WFR flow** of MMD. It works only slightly worse than IFT but does not come with guarantees. See our overall rebuttal summary and the PDF.
>
> > Proposition 3.8 immediately follows from two well-known facts... for the KL case, it is not a surprising property ...
>
> We have already clearly stated that it is standard (L229-230). Neither did we claim it is surprising. It is also immediately evident that KL is not the focus of this paper, as can be seen from the shortness of Sec.3.3. Hence, we believe our current presentation is not likely to cause confusion. We emphasize that the discussion of KL is for completeness and to show that IFT enjoys the *best of both worlds*.
>
> Bonus: Prop. 3.8 is not as trivial as it seems -- we just fixed an error in the current revision: the LSI only holds along the mass-preserving spherical flows (SIFT) over $\cal P$, but the flow over $\cal M ^+$. This technicality does not change ML applications, especially with MMD. But anyway, thanks for helping us notice this.
>
> > For the MMD case, the results are quite novel... other works... (Arbel et al., 2019). Thus, I am wondering, are the provided proofs of exponential convergence rates actually important...?
>
> Thank you for the positive assessment. We will add a small discussion in the revision as per your comment. A sketch:  (Arbel et al., 2019) does not contain (global) convergence analysis. For example, their Prop.2 states energy is non-increasing. However, this is not equivalent to convergence and is easily satisfied by many flows. The mathematical limitation to their "MMD flow" is that the MMD is in general not guaranteed to be convex along the Wasserstein geodesics. [Arbel et al.] also contains a heuristic noise injection procedure, albeit without good analysis characterization. In contrast, we believe establishing the first global (exponential) convergence in our paper is, needless to say, important. Not to mention our theorems/proof are clean and do not contain un-verifiable assumptions. Therefore, in our case, we have both practical performance and theoretical guarantees
>
> > Does your approach have some practical use cases?
>
> This paper is mainly methodological and analytical, but we also value practical application. MMD flow applications in the literature mainly include, e.g.,
>
> - image process, as done by [Hertrich et al.] and that research group at TU Berlin
> - generative models (MMD GAN)
> - MMD two-sample test
>
> We also see some new directions of new deep generative models that are based on gradient flows of the MMD e.g. in [Galashov et al.]
>
> > Strengths: The paper provides the proof of...
>
> We emphasize that the proofs are a part of but not the whole of our contributions, which also include the discovery of the IFT **gradient structure**, the implementable algorithms, PDE theory of the dissipation mechanism of the MMD and inf-convolution with Wasserstein (e.g. Prop 3.2, Thm 3.6). The work also advances the state-of-the-art understanding of "MMD gradient flow", which already has sizable literature.
>
> ## Reference
> Galashov A, de Bortoli V, Gretton A. Deep MMD Gradient Flow without adversarial training. arXiv; 2024

---

> ### Author Response · Authors · 2024-08-12
> **End of discussion period approaching**
>
> Dear reviewer,
>
> Thank you for your feedback on our manuscript. We have carefully considered your comments and suggestions and have made the revisions. Per your suggestions, we have also included new experiments and done our best to answer your questions.
>
> As the discussion period will be closed soon, we wish to ask for your feedback on the rebuttal kindly. Have we addressed your concerns? Is there anything else we can improve?
>
> Thank you again for your time and effort.
>
> Authors

---

> > ### Comment · Reviewer_2Ho7 · 2024-08-13
> >
> > I thank the authors for their answers to my questions and concerns. First, I appreciate that you conduct a moderate-dimensional (d=100) experiment with mixture of 3 Gaussians as a target and include the comparison with some of the requested approaches. I expected that you will also provide some figures visualizing the obtained results (although, I understand that it might be quite tricky for d>2). Second, as you explain, the practical implementation of KL energy with the proposed IFT flow (requested by me and other reviewer) is out of the scope of this paper. From my point of view, you should somehow state this directly in the paper, otherwise, the existence of the whole section of the paper related to this case seems to be confusing from my point of view. Third, I am still not sure that the provided theoretical results with only moderate-dimensional experiments (up to d=100) with Gaussians are solid enough to be published. I see that the limited experimental evaluation ('proof-of-concept' type of experiments) was noted by other reviewers too.
> >
> > Respecting the time spent by the authors on running the experiments, I *adjust my score* accordingly. Meanwhile, I am looking forward for further discussion with other reviewers and Area Chairs.

---

> > > ### Author Response · Authors · 2024-08-14
> > > **Thank you for responding to our rebuttal**
> > >
> > > Dear reviewer,
> > >
> > > Thank you for considering our rebuttal. We appreciate your feedback and that you have adjusted your score accordingly.
> > >
> > > We agree and will indeed provide a detailed explanation of the sampled-based (MMD energy) vs score-based (KL energy), as outlined in our rebuttal text, especially around Sec 3.3.
> > >
> > > We respect the reviewer's third point. Indeed, our experiments may be "proof of concept". Our hope is to propose the IFT "gradient structure" in this paper (e.g. K_IFT in eq (7) ) and study its properties (e.g. Theorem 3.6). In view of the already sizable literature on MMD flows started by Arbel et al. (2019), we believe that the proposed gradient structure is a significant contribution and will generate useful mathematical insights for ML researchers working on related topics. From the technical perspective of gradient flows, the discovery of a new gradient structure is already quite non-trivial. That is our intention in this paper. However, we perfectly respect that the reviewer may have a different perspective based on their expertise. We will do our best in the next revision to make our insight useful for a wider audience.
> > >
> > > Thanks again,
> > > Authors

---

### Official Review · Reviewer_rKJj · 2024-07-13

**Soundness:** 3
**Presentation:** 4
**Contribution:** 4
**Rating:** 6
**Confidence:** 2

**Summary:**

This paper proposes a novel gradient flow geometry (IFT), based on the infimal convolution of the Wasserstein tensor with the MMD tensor. For this geometry, the authors show global exponential convergence guarantees for both MMD and KL energies. They then develop an algorithm for the IFT gradient flow and test it on an MMD inference task, showing empirically that it avoids mode collapse as Arbel et al. does.

**Strengths:**

1. The exposition clarity is excellent. The authors do a great job of positioning their work relative to existing gradient flow works.
2. In introducing a novel gradient flow geometry and showing favorable convergence characteristics, the work has good potential to inspire follow-on works.

**Weaknesses:**

1. The experiments run are relatively simple and low-dimensional, and it is not clear how practical the method would be for more realistic application scenarios.
2. There is no example comparison of behavior for KL, which would have been nice to see.

**Questions:**

1. Looking at the numerical example, have you tried any heuristic approaches, e.g. some sort of branching, for eliminating and repopulating particles when weights get very low on certain particles? It seems such behavior might improve performance empirically.

**Limitations:**

Limitations are well-acknowledged by the authors.

---

> ### Author Rebuttal · Authors · 2024-08-04
>
> We thank the reviewer for the fair assessment and constructive suggestions. We are glad that the exposition of the paper is easy to follow -- thank you for this feedback. We have incorporated some of your suggestions and left more challenging ones for future work.
>
> > experiments run are relatively simple and low-dimensional, and it is not clear how practical
>
> Your assessment is correct -- indeed our experiments are only a proof-of-concept. We have now included results in higher dimensions in the attached PDF. We also plan to investigate more realistic applications in the future.
>
> > numerical example, have you tried any heuristic approaches,
>
> We appreciate your constructive suggestions.
>
> First, we stated at the end of Sec.5 (L317-319) that there are simple improvements that can be made to improve the algorithm, and we deliberately left them out to keep the paper focused; almost playing handicapped. The IFT still outperforms the pure Wasserstein flow.
>
> Second, thank you for the valuable suggestions. We believe they can definitely help with practical performance. As we reported in the paper, we actually let IFT run with a handicap (i.e. every IFT iteration counts as two iterations). We did so to show that IFT already has an advantage without further heuristic tricks. We have focused on implementing the WFR flow during the rebuttal to provide another new algorithm for comparison. We will further investigate your suggestions in the near future and comment in our Sec.5 if any of the tricks help.
>
> > comparison of behavior for KL
>
> We agree that more experiments can strengthen the paper. A direct comparison between energy MMD vs KL is difficult, as we explain below. However, we did run the newly implemented WFR flow and reported them in the attached PDF.
>
> First, We must note that KL minimization is NOT the topic of this paper: our focus is a type of (unbalanced-transport) gradient flows and optimization with applications to the MMD-minimization tasks. IFT is tailored to address the gap in the literature on "MMD flow". We also do **not** claim that MMD inference is superior to KL inference, while some other papers might do so. Furthermore, we look forward to applying our theory to generative models using MMD, e.g., [Galashov et al].
>
> Nonetheless, we appreciate the reviewer raising this. We can think of two interpretations of the question:
>
> 1. KL energy with the proposed IFT flow:
> Recall that the MMD step is a discretization of the differential equation
> $$
> \dot \mu =- \beta\cdot \operatorname{\mathcal{K}}^{-1}
> \left(\log \frac{\mathrm{d}\mu}{\mathrm{d}\pi} - \frac{\int \operatorname{\mathcal{K}}^{-1}\log \frac{\mathrm{d}\mu}{\mathrm{d}\pi}}{\int \operatorname{\mathcal{K}}^{-1}1}\right)
> $$
> One could view the update step as a step in the "kernel-mean-embedding space": let $e(\mu):=\operatorname{\mathcal{K}} \mu$  be the kernel mean embedding of $\mu$, then the update rule is
> $$
> e(\mu)^{\ell+1}\gets e(\mu)^{\ell}  - \eta \beta \cdot \left(\log \frac{\mathrm{d}\mu^{\ell}}{\mathrm{d}\pi} -...\right)
> $$
> However, as there is no guarantee that the velocity $\log \frac{\mathrm{d}\mu^{\ell}}{\mathrm{d}\pi}-...$ to be in the RKHS. Hence, this step is theoretically interesting, but it's unclear how the infinite-dimensional update can be implemented in a principled way.
>
> 2. KL energy with the WFR flow: this case belongs to the "KL-inference" where one only has access to the score (or potentials) of the target e.g. $\nabla \log \pi$, instead of the "MMD-inference" (our paper, and Arbel et al., etc.) where we have access to the target $\pi$ via its samples but not the score $\nabla \log \pi$. Therefore, how to perform a "fair comparison" is unclear to us at the moment (e.g. how many samples from $\pi$ ? Noisy evaluation of the score?).
>
> In summary, we agree that the reviewer's proposal is interesting and can see room for future work. At this moment, we can only leave comparing MMD vs KL as future work as they apply in different tasks and settings (score-based vs sample-based).
>
> We have now implemented the **WFR flow** of MMD. It works only slightly worse than IFT but does not come with guarantees. See our overall rebuttal summary and the PDF.
>
>
> ### Reference
> Galashov A, de Bortoli V, Gretton A. Deep MMD Gradient Flow without adversarial training. arXiv; 2024

---

> ### Author Response · Authors · 2024-08-12
> **End of discussion period approaching**
>
> Dear reviewer,
>
> Thank you for your feedback on our manuscript. We have carefully considered your comments and suggestions and have made the revisions. We have also included new experiments and done our best to answer your questions.
>
> As the discussion period will be closed soon, we wish to ask for your feedback on the rebuttal kindly. Have we addressed your concerns? Is there anything else we can improve?
>
> Thank you again for your time and effort!
>
> Authors

---

> > ### Comment · Reviewer_rKJj · 2024-08-12
> > **Thank you & keeping score as is**
> >
> > Dear authors,
> >
> > Thank you for the clear and extensive response to my review. I think I will maintain my score as is, remaining slightly positive on the paper, given the proof-of-concept experiments and perhaps limited immediate impact.
> >
> > Reviewer

---

> > > ### Author Response · Authors · 2024-08-14
> > > **Thank you for reading our rebuttal**
> > >
> > > Dear reviewer,
> > >
> > > Thank you for your feedback and for taking the time to read the rebuttal.
> > >
> > > Best regards,
> > > Authors

---

### Author Rebuttal · Authors · 2024-08-06

Dear reviewers, dear AC,

We would like to thank all reviewers for their constructive feedback. We are glad that the majority of the reviewers found our paper easy to read and contains non-trivial contributions.

The major concern expressed by some reviewers is that more experiments would strengthen the paper -- which we have now done.

Based on the reviewers' suggestions, we have included new experiments. The figures are reported in the attached PDF. For example,

1. experiments in higher dimensions $d>2$;
2. a new Wasserstein-Fisher-Rao gradient flow of the MMD energy; see the appendix below for details. This is the first implementation of the MMD energy in this flow to the best of our knowledge. Note that this is the same algorithm used in [Yan et al. & Lu et al.], as requested by one reviewer. As we are not in the score-based sampling regime, we have to use the MMD energy instead of the KL energy (mentioned in some reviews);
3. many other improvements in terms of scalability and practicality, as suggested by the reviewers.

Therefore, we believe the reviewer's concerns have been addressed. We hope the reviewers will take our new results and improvement into consideration. If there are any further concerns, we are happy to discuss them during the discussion phase.

Thank you for your time and consideration,
Authors



## Appendix: Details on the newly implemented WFR flows for comparison (i.e., KL step for reweighting)

In the second paragraph in Sec.3.4 (L248-256), we discussed the comparison with the Wasserstein-Fisher-Rao (WFR) flow of the MMD. Note that there are no sound convergence guarantees for this scheme yet. To put things in perspective, let us make it clear: Yan et al. and Lu et al. use variants of WFR flows, but with KL energy. We can not directly compare with them as we are in the sample-based regime (only have access to samples of $\pi$) and they are in the score-based regime (only have access to the score $\nabla \log \pi$).

Nonetheless, per the reviewer's suggestion to compare with them, we have implemented the **WFR flow of the MMD energy**. Note that this is the first implementation of the MMD energy in this flow, to the best of our knowledge. We believe this has added value to the paper and to some extent addresses the reviewer's demand for more empirical comparisons. This amounts to the JKO splitting steps
$$
            \mu^{\ell+\frac12}
            \gets\arg\min_{\mu\in\cal P} F(\mu ) + \frac1{2\tau}W_2^2(\mu, \mu^\ell)
            \textrm{(Wasserstein step)}
  $$
$$
            \mu^{\ell+1}
            \gets\arg\min_{\mu\in\cal P} F(\mu ) + \frac1{\eta}{\mathrm{KL}}(\mu, \mu^{\ell+\frac12})
             \textrm{(KL step)}
$$
As is in the Wasserstein step, in practice, we use the
explicit Euler scheme which boils down to the **entropic mirror descent**.
As is well-known, especially in the optimization literature, the entropic mirror descent step can be implemented as multiplicative update of the weights (or density), i.e.,
suppose $x_i^{\ell+1}$ is the new particle location after the Wasserstein step, then we update the weights vector $\alpha$ via
$$
            \alpha_i ^{\ell+1}
            \gets \alpha_i  ^{\ell} \cdot \exp
            \left(
                -\eta \cdot
                \frac{\delta F}{\delta \mu}[\mu^\ell] (x^{\ell+1}_i)
            \right)
$$
where the Riemannian velocity $\frac{\delta F}{\delta \mu}[\mu^\ell]$ is the same as used in the Wasserstein step above. For the MMD, it is given by (already provided in the manuscript)
$$\frac{\delta F}{\delta \mu}[\mu^\ell]=\int
k(x, \cdot )
(\mu^\ell -\pi  )(\mathrm d x)$$

We have added more details to the appendix of the revised manuscript. Numerical results are reported in the PDF below.

---

### Decision · Program_Chairs · 2024-09-25

**Decision:**

Accept (poster)

**Comment:**

This manuscript proposes a new gradient flow over probability and non-negative measures termed the interaction-force transport (IFT). The flow is based on the inf-convolution of the Wasserstein Riemannian metric tensor and the spherical maximum mean discrepancy (MMD) Riemannian metric tensor. The authors provide a number of theoretical results related to their proposal: global exponential convergence guarantees for the MMD and Kullback-Leibler divergence energies. While convergence results are available for the KL divergence energy, not much is known theoretically in the context of the MMD energy. The authors then introduce a particle gradient descent algorithm for IFT, composed of two steps gradient flows (one to update particle locations via the Wasserstein step and one to update particle weights via the MMD step), and show two proof-of-concept examples with 2D Gaussians and Gaussian mixture that validate the developed theory.
In the rebuttal, the authors conducted higher dimensional experiments and implemented WFT flow for comparison, which enhance the work.

The paper is well-written and well-organized. Despite heavy notation throughout, the authors do a nice job introducing the notation and staying consistent throughout the manuscript. The literature review on related method is also quite thorough. This work has novelty and significant theoretical contributions to the machine/statistical learning literature:  the new definition of gradient flows, the proofs of global convergence for the MMD and KL energies, and comparison to previously defined methods for the MMD flow that required a heuristic noise injection step. While the paper is theoretical in nature, the authors also provide an implementation of the proposed gradient flows. The presented algorithm is easy to understand ensuring reproducibility. The topic is timely and will appeal to a broad NeurIPS audience.

The work has great theoretic contribution and has the potential to make impacts in the field. I recommend the acceptance.